# Viviparity imparts a macroevolutionary signature of ecological opportunity in the body size of female *Liolaemus* lizards

Saúl F. Domínguez-Guerrero [1] ✉, Damien Esquerré[2], Edward D. Burress [1,4], Carlos A. Maciel-Mata [3], Laura R. V. Alencar [1] & Martha M. Muñoz[1]

Viviparity evolved ~115 times across squamate reptiles, facilitating the colonization of cold habitats, where oviparous species are scarce or absent. Whether the ecological opportunity furnished by such colonization reconfigures phenotypic diversity and accelerates evolution is unclear. We investigated the association between viviparity and patterns and rates of body size evolution in female *Liolaemus* lizards, the most species-rich tetrapod genus from temperate regions. Here, we discover that viviparous species evolve ~20% larger optimal body sizes than their oviparous relatives, but exhibit similar rates of body size evolution. Through a causal modeling approach, we find that viviparity indirectly influences body size evolution through shifts in thermal environment. Accordingly, the colonization of cold habitats favors larger body sizes in viviparous species, reconfiguring body size diversity in *Liolaemus*. The catalyzing influence of viviparity on phenotypic evolution arises because it unlocks access to otherwise inaccessible sources of ecological opportunity, an outcome potentially repeated across the tree of life.

Viviparity (live-bearing) is a major evolutionary innovation that affords enhanced maternal control over embryonic development, and drastically reshapes how organisms interact with their thermal habitat[1–3]. Through active thermoregulation, viviparous females control the incubation temperature for developing embryos, optimizing the neonate's phenotype, maximizing offspring survivorship, and influencing offspring sex[4–7]. Innovations are theorized to create ecological opportunity by allowing evolutionary access to new niches, in turn catalyzing phenotypic evolution and increasing diversification rates[8]. Viviparity is thought to enhance access to cold environments: whereas the species richness of oviparous species decreases with environmental temperature (substrate temperature in cool habitats may constrain embryogenesis of eggs), the percentage of viviparous species increases at high latitudes and/or elevations (where a warmer female temperature, relative to environment, enhances embryonic development)[9–13]. Independent transitions to

similar environments are predicted to drive convergent phenotypes[14,15]. Further, transitions into uncolonized habitats such as islands, lakes, or sky islands like mountaintops are predicted to release organisms from ancestral competitors and/or predators, in turn accelerating evolution[16,17]. Colonization of East African lakes by cichlids or Caribbean islands by anoles, for example, promoted convergent morphologies and/or fast phenotypic diversification associated with niche partitioning[18,19]. Most transitions to live birth in vertebrates cluster in squamate reptiles (lizards and snakes), with at least 115 independent origins identified[20]. Despite rampant convergence in the evolution of this reproductive innovation, we do not know if independent transitions are predictably associated with phenotypic convergence and/or increases in the rate of phenotypic evolution. Correspondingly, despite strong conceptual support, the empirical bridge linking this innovation to ecological opportunity and trait evolution remains uncertain.

¹Department of Ecology and Evolutionary Biology, Yale University, 06511 New Haven, CT, USA. ²School of Earth, Atmospheric and Life Sciences, University of Wollongong, Wollongong, NSW 2522, Australia. ³Predio Intensivo de Manejo de Vida Silvestre X-Plora Reptilia, 43350 Metztitlán Hidalgo, México. ⁴Present address: Department of Biological Sciences, University of Alabama, Tuscaloosa, AL 35487, USA. ✉e-mail: saul.dominguezguerrero@yale.edu

As body size is often associated with life history traits, including viviparity[21-23], we centered this study on body size to test the potential association between parity mode and evolutionary phenotypic shifts. Crucially, however, the pathways linking parity mode and body size evolution are potentially complex and imbricating. Although viviparity can be associated with shifts in body size in chondrichthyans[23], live birth is simultaneously associated with low environmental temperatures in squamates[13,24], which is a good predictor of larger body sizes in endotherms (Bergmann's rule[25-27]) and some ectothermic vertebrates[28-30]. Low environmental temperatures tend to be, in turn, associated with the use of saxicolous (i.e., rocky) substrates for thermoregulation[31,32] and with certain diet preferences[33], two features also recognized as drivers of shifts in animal body size[34,35]. In short, there is multiplicity and non-independence of extrinsic and intrinsic features that can shape body size evolution, which presents a conundrum: to what factor (or interaction among factors) does body size evolution respond? Therefore, we also were interested in exploring whether the focal innovation, live birth, has a direct evolutionary effect on body size, or whether body size is associated with shifts in other attributes that are correlated with viviparity (i.e., non-intrinsic effects of live birth on body size). Ideally, a study integrating environmental temperature, substrate use, and diet would focus on a lineage characterized by independent transitions in parity mode (naturally replicated framework), and in which all these traits vary among species.

Liolaemidae is South America's most species-rich lizard radiation[36], with 341 described species in three recognized genera: *Ctenoblepharis*, *Phymaturus*, and *Liolaemus*[37]. *Ctenoblepharis* is an oviparous, monospecific genus and *Phymaturus* (52 spp) is a fully viviparous genus. *Liolaemus* (289 spp), by contrast, is characterized by numerous transitions in parity mode[36,38]. *Liolaemus* is known for markedly high rates of body size evolution (snout-vent length; SVL)[39] and, as a group, occur in every habitat in South America, from sea level to more than 5400 meters in elevation, and from tropical to temperate latitudes (from −9.5 to −54°S)[36,40]. The geographic distribution of this lineage is truly extraordinary: *Liolaemus* includes the world's highest-elevation lizard (*L.* aff. *tacnae*)[40], and the southernmost lizard species in the world (*L. sarmientoi* and *L. magellanicus*)[41]. As in many other squamate groups, viviparous *Liolaemus* species inhabit cold environments and the substrate choice of the lineage is well characterized[36,42,43]. Furthermore, *Liolaemus* is characterized by multiple transitions in diet (insectivory, omnivory, and herbivory)[33,44]. Putting these features together, *Liolaemus* represents a useful group with which to test if live birth is predictably associated with shifts in

body size evolution, and to disentangle the evolutionary relationships among environmental temperature, diet, and substrate choice shaping the viviparous phenotype.

Here, we use a series of phylogenetic analyses to test: (1) the association between parity mode and evolutionary shifts in body size (evolutionary optimum, and evolutionary rate), and (2) if viviparity influences directly and/or indirectly (by interaction with other ecological features) body size evolution in *Liolaemus* lizards. We discover that viviparity in *Liolaemus* is strongly associated with larger optimal body sizes (estimated under an Ornstein-Uhlenbeck model of adaptive evolution), but not with faster body size evolution. The fact that viviparous species are larger than oviparous species arises because live birth potentiates access to cold environments, and low environmental temperatures, in turn, drive larger body sizes. Together, our results indicate that viviparity facilitates colonization of novel habitats, in turn exposing organisms to new selective pressures and driving convergent phenotypic evolution, a synergism that may be repeated across transitions to live birth – and different innovations – in other lineages.

## Results

We gathered data on adult body size (SVL), parity mode, diet, substrate use, and Mean Annual Temperature (MAT) from 133 species (58 oviparous and 75 viviparous) of *Liolaemus* lizards (Supplementary Data 1). Body size in our *Liolaemus* data varied more than two-fold among species, ranging from 40 mm to 100 mm (Supplementary Fig. 1). We found that optimal body size is larger in viviparous than oviparous species (θ = 66.1 mm and 56 mm, respectively; Fig. 1a; Supplementary Table 1), a pattern robust to phylogenetic uncertainty (Supplementary Table 4). When testing whether diet or substrate use are associated with different patterns of body size evolution, we found that insectivorous and omnivorous/herbivorous species share an optimal SVL (θ = 61 mm, Supplementary Tables 2, 5) and that saxicolous lizards exhibit a larger optimal body size than their terrestrial counterparts (θ = 66.3 mm and 57.3 mm, respectively; Fig. 1b; Supplementary Table 3, 6). When we simultaneously compared models that consider parity mode, diet, or substrate use as predictors of body size evolution, we found that the two optima, single-rate model of parity mode − where viviparous species evolve a larger optimal body size than their oviparous relatives− was the best supported (Supplementary Tables 7, 8). Patterns of body size evolution in *Liolaemus* seem to be more influenced by parity mode than by substrate choice. The strength of convergence in body size, however, was weak among viviparous

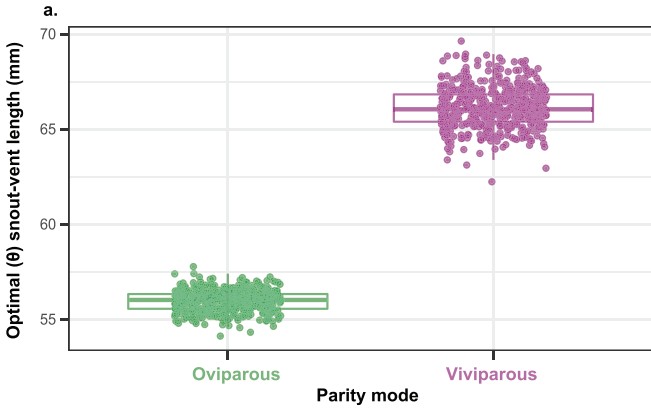

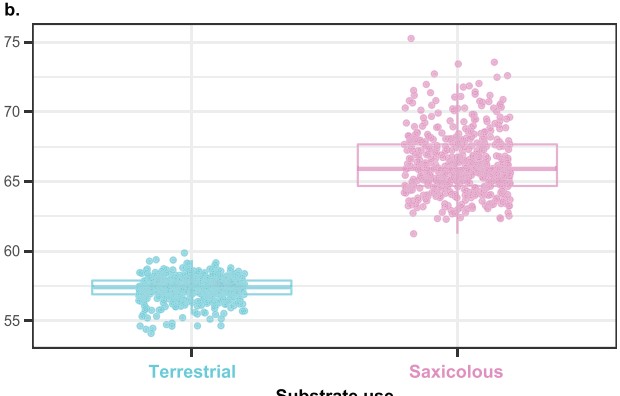

**Fig. 1 | Parity mode (a) and substrate use (b) predict optimal (θ) body size shifts in *Liolaemus* lizards. a** Viviparous (purple) species have a larger optimal (θ) body size than oviparous species (green). **b** Saxicolous (pink) species have a larger optimal (θ) body size than their saxicolous (blue) relatives. Evolutionary optimal body size was inferred from an Ornstein-Uhlenbeck (OU) model-fitting procedure (see

Methods). The plots display the distribution of the estimated optimal body sizes, where the line is the median, box indicate lower and upper quartiles, and whiskers are minimum and maximum values. Each point within the plots represents a different stochastic character map (*n* = 500) across the ultrametric tree. Source data are provided as a Source Data file.

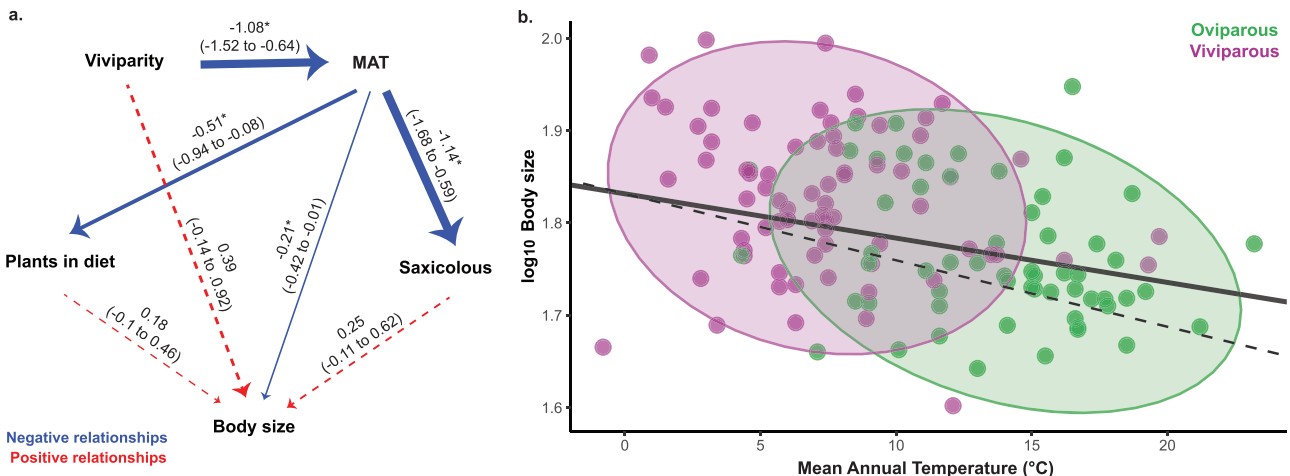

**Fig. 2 | Changes in the thermal environment drive body size evolution in *Liolaemus* lizards. a** Average of the five best-fitting models show that viviparity has an indirect effect (through its association with mean annual temperature (MAT)) on the body size of *Liolaemus* lizards. Solid arrows (and asterisk) represent significant associations and dashed arrows denote relationships that were not significant. Thicker arrows indicate stronger effects. Within parentheses we show the confidence interval of each association. This average model was supported from a Phylogenetic Path analysis (Supplementary Table 10), and the confidence intervals were obtained from 500 bootstrap replicates. **b** At lower environmental temperatures, *Liolaemus* lizards tend to be larger. Viviparous species (purple) are larger and common in colder habitats than their oviparous (green) counterparts. Solid line represents evolutionary regression ($\log_{10}$ body size = −0.0048*MAT + 1.8316, $n = 132$) and dashed line represents optimal regression ($\log_{10}$ body size = −0.0072*MAT + 1.8316, $n = 132$). Both regressions (evolutionary and optimal) were simultaneously estimated in an Ornstein-Uhlenbeck framework (Supplementary Table 11). Source data are provided as a Source Data file.

species ($w = 1.007$, $p = 0.1$, $n = 133$) and among saxicolous lizards ($w = 0.892$, $p = 0.8$, $n = 123$). Although viviparous and oviparous species evolve towards different optimal body sizes, we found that the rate of this SVL evolution was comparable among parity modes (Supplementary Table 9; posterior probability (PP) that the rates are state-dependent = 0.3–0.67). Similar rates of body size evolution are maintained even when comparing species by diet or substrate use (Supplementary Table 9). In short, parity mode, diet, or substrate use do not accelerate rates of body size evolution in *Liolaemus* lizards.

As viviparous species (which are primarily saxicolous)[43] evolve larger optimal body sizes than oviparous species, we were interested in evaluating whether live birth is directly or indirectly associated with body size shifts. When we tested the different hypotheses about the direct, indirect, or direct/indirect effects of viviparity on body size (Supplementary Fig. 2), we found that models 11, 13, 14, and 15 (in which viviparity is directly and indirectly associated with body size) and model 8 (in which live birth indirectly influences body size) were equally well supported (Supplementary Table 10). Then, we performed a model averaging[45,46] of the five best-fitting models (Fig. 2a). The average model indicates that live birth indirectly (but not directly) influences body size through its association with environmental temperature. When viviparity evolves, species tend to occupy colder environments, and their presence in regions with low environmental temperatures simultaneously influence preferences for rocky perches, ingestion of plants in the diet, and larger body sizes (Fig. 2a). Transitions from terrestrial to saxicolous substrates, or from insectivorous to omnivorous/herbivorous diet, however, are unrelated to body size shifts (Fig. 2a). Lastly, we found that the optimal regression between body size and air environmental temperature under OU models is decoupled from evolutionary regression (Fig. 2b; Supplementary Table 11). As phylogenetic half-life bounded away from 0, it indicates a lag in body size adaptation to environmental temperature. While body size does adapt to the thermal environment, it does so rather slowly, in turn reflecting a relatively long phylogenetic half-life ($t_{1/2}$) of 7.3 million years (28% of total tree length) for oviparous species and of 5.7 million years (22% of total tree length) for viviparous species, and a rate of adaptation ($\alpha$) of 0.1 and 0.12, respectively (Supplementary Table 11).

## Discussion

When similar habitats are independently colonized, organisms often evolve convergent phenotypes[19]. Likewise, colonization of undersaturated habitats reduces predation and interspecific competition, while simultaneously driving increases in population densities, and fast phenotypic diversification by niche partitioning[8,47]. For example, the return to sea from land by ichthyosaurs and cetaceans is associated with evolution of larger sizes and rapid body size diversification, a product of abundance of food and trophic specialization[35]. Here, we found that viviparous *Liolaemus*, which repeatedly colonized cold habitats in South America[12,36], exhibit larger sizes but similar rates of body size evolution than oviparous species. Comparing different causal paths, we discovered that viviparity only weakly influences body size evolution directly. Instead, live birth is strongly associated with the ability to inhabit cold habitats, and low environmental temperatures, in turn, are associated with shifts to larger body sizes. Below, we describe some potential mechanisms (which are not mutually exclusive) behind the patterns of body size evolution observed in this study.

Parity mode and substrate use are good predictors of body size evolution in *Liolaemus* lizards (Fig. 1) and other animal species[23,34,48]. Yet, parity mode dictates body size shifts more strongly than substrate use in our system (Supplementary Tables 7, 8). Although live birth is associated with larger body sizes in *Liolaemus*, we discovered that it occurs because viviparous species inhabit cool environments (Fig. 2a). Uplift of the Andes took place over the last ~100 million years and the origin of *Liolaemus* likely occurred ~26 million years ago (late Paleogene) in the Central Andes[36,49,50]. From there *Liolaemus* dispersed into other regions of South America, facilitated by climatic and geological changes and evolution of live birth[36,42]. Beginning in the early Miocene (~20 MY ago), *Liolaemus* colonized Patagonia[51] where environmental temperature has decreased through time[52]. Alternatively, *Liolaemus*, or at least some major clades (such as the *Liolaemus* subgenus), could have originated in Patagonia and from there dispersed into other South American regions[49,53,54]. Since the origin of *Liolaemus* (independently of the ancestral geographic origin), the Andes continued to rise, shaping new thermal habitats at high elevation[55]. Whereas the rise of the Andes restricted oviparous species mainly into warmer lowlands, independent transitions from oviparity to live bearing underpinned

independent colonization of cold habitats (into high elevation and/or latitude)[12,36]. Thus, during their evolutionary history, viviparous *Liolaemus* experienced colder environmental temperatures across space and time[52]. As viviparous species were colonizing high-latitude habitats, and as the Andes mountains became higher, they were exposed to progressively colder thermal habitats[52]. Because environmental temperature is a direct predictor of body size evolution in *Liolaemus* (Fig. 2), the continuous changes of environmental temperatures over million years could have driven the larger sizes of viviparous species.

Since Bergmann's foundational work nearly two centuries ago, we have recognized that animals (such as birds, mammals, and some reptile species) in colder habitats tend to be larger than relatives in warmer habitats (Bergmann's rule), likely as a response to maintain a stable body temperature[25–27,30]. Consistent with Bergmann's rule, larger *Liolaemus* species are found in colder habitats (Fig. 2b). Larger body sizes in cool habitats were previously documented in liolaemid lizards and may afford an advantage for heat balance (larger species show slower heating and cooling rates)[56–59]. Yet, the validity of Bergmann's rule has been debated in *Liolaemus* and other ectothermic vertebrates[60–65]. Furthermore, the optimal regression between body size and environmental temperature is still decoupled from the evolutionary regression (Supplementary Table 11), indicating that factors besides environmental temperature likely also contribute to body size shifts[66,67]. Alternatively, temperature during early development (temperature-size rule[68,69]) could play an important role in body size evolution. At lower developmental temperatures individuals grow more slowly but reach larger adult body sizes than individuals developing at higher temperatures[68–72]. If viviparous neonates of *Liolaemus* are exposed to lower developmental temperatures than oviparous neonates (a topic poorly explored), then it could explain the differences in body size between parity modes.

In addition to effects due to temperature (environmental or developmental), release from predators and greater food availability could be associated with larger body sizes in *Liolaemus*. Colonization of novel habitats, where predators are scarce and/or there is high food availability, is associated with larger sizes across squamates[73]. Cool habitats at high latitude and/or elevation are characterized by lower predatory risk for lizards[74–76]. In those cold environments, reptiles have a lower extrinsic mortality (predation) and greater longevity than species from warm habitats[77]. Indeed, the same work cited above includes the maximal longevity for four viviparous and two oviparous *Liolaemus*, and although the sample size is too small to perform a statistical analysis, viviparous species are evidently more long lived than oviparous species (13 years vs 6 years, respectively)[77]. Therefore, it is possible that viviparous *Liolaemus* from cold habitats have higher survival rates than their oviparous relatives from warm habitats, driving the evolution of larger body sizes. Likewise, as the high-elevation Andes are cradles of plant and insect diversity[78–80], it is possible that viviparous lizards can consume more food items than oviparous species, and access to greater resources may favor the evolution of larger sizes. Further, compared with their oviparous relatives, viviparous *Liolaemus* exhibit greater activity hours[43], which may afford more time for foraging and confer more energy available for growth. Lastly, viviparous females could be larger than their oviparous relatives as an advantage to have more abdominal space for the developing embryos and/or to produce more neonates[81]. However, these hypotheses have been tested and not supported in *Liolaemus*[81–83]. Together, over-representation in cool habitats (where larger body sizes likely convey thermoregulatory advantages, or are a product of low developmental temperatures), release from predators, and high food availability, could all interact to drive the larger body sizes observed in viviparous *Liolaemus*.

Although colonization of cold habitats by viviparous *Liolaemus* is predictably associated with evolution of larger body sizes (Fig. 2), the rate at which body size evolves is comparable between oviparous and viviparous species (Supplementary Table 9). Even when we compared rates of body size evolution among species by diet or substrate use, we did not identify differences in the tempo of SVL diversification (Supplementary Table 9). Ecological opportunity often results in a shift in the tempo or mode of phenotypic evolution, but uncommonly both. For example, habitat transitions from rivers to lakes in cichlid fishes accelerates phenotypic diversification, but not in expansion to novel phenotypic space[84]. We suggest that similar rates of body size evolution in oviparous and viviparous *Liolaemus* could be related (but not limited) to stabilizing selection, a lack of ecological release from competitors for viviparous species, or release of competitors and selection in both parity modes. We consider these in more detail below.

Colonization of cold habitats by viviparous ectotherms (such as *Liolaemus*) depends on the ability of individuals to regulate their field body temperature within their preferred range (i.e., thermoregulatory behavior) and maximize their performance for fitness-based activities[3,85,86]. Indeed, behavioral thermoregulation in viviparous *Liolaemus* (which are mainly saxicolous) is more precise than in their oviparous relatives[85,86]. In cold habitats, rocks and boulders afford warmer and more stable temperatures than the ground, bushes, or trees, making them ideal substrates for behavioral thermoregulation[31,32]. For example, at high latitude (51°S) *Liolaemus sarmientoi* perch on rock promontories and exhibit warmer and stabler field body temperatures (within their preferred range) than *L. magellanicus* (a sand-dwelling species) even though both species are viviparous[41]. Therefore, use of saxicolous substrates underpins transitions from warm to cool habitats in viviparous *Liolaemus* (Fig. 2)[43]. The need for suitable nesting sites could restrict the use of rocky environments for oviparous species, but nesting constraints will not limit habitat use in viviparous species[82]. Although the use of rocks for thermoregulation (and crevices for refuge) has been considered as a driver of faster rates of body size evolution in monitor lizards[34], it has also been proposed as a mechanism to explain morphological stasis in *Phymaturus* lizards (sister lineage of *Liolaemus*)[87]. Similar to *Phymaturus*, it is possible that a strong dependence on rocks and crevices in viviparous *Liolaemus* is associated with stabilizing selection for an adaptive optimum[87]. In other words, rapid changes in body size could constrain the thermoregulatory performance space of viviparous species. Alternatively, rocky substrates, which are ideal for structural partitioning (smaller individuals perching on small rocks and larger lizards occupying large rocks[88–90]), could have been occupied by other viviparous squamate competitors during the colonization of cold habitats by viviparous *Liolaemus*.

*Phymaturus* is a genus comprised of viviparous and saxicolous lizards from cold habitats, and is the sister lineage to *Liolaemus*[36]. Most transitions from oviparity to viviparity in *Liolaemus* are predicted to have occurred after the origin of *Phymaturus*[36,38]. Therefore, it is possible that when viviparous *Liolaemus* colonized cool environments there was not a complete ecological release of antagonists because *Phymaturus* lizards were already inhabiting cold habitats and rocky substrates[36,38]. In other words, evolution of live birth and colonization of cool habitats in *Liolaemus* released viviparous species from their egg-laying relatives, but not from other live-bearing competitors. Thus, low environmental temperatures promoted larger sizes in viviparous species, but there was not opportunity for rapid body size evolution associated with structural partitioning. Indeed, evidence of body size partitioning in *Liolaemus* is not supported among species or between sexes[91]. Lastly, it is important to note that the Andes is a cradle of several evolutionary radiations where plants and animals (including liolaemids) have rapidly diversified. The Andes is one of the most topographically complex regions around the world, characterized by a wide variety of climates and habitats that represents a continuous source of ecological opportunity for lizards[52,92]. As such, it is possible that we did not find signatures of rapid body size shifts in viviparous

species because the *Liolaemus* genus as a whole is characterized by rapid body size evolution.

In conclusion, we find that live birth, a reproductive innovation, promotes the evolution of larger body sizes in *Liolaemus*. Yet, the effect of live birth in body size shifts is mediated by environmental temperature. Larger sizes in viviparous species reflect the interaction between viviparity and other sources of ecological opportunity, namely access to cold habitats. Innovations are hypothesized to open access to new adaptive zones (sensu Simpson[93]), in turn prompting phenotypic evolution. Here we found that the effect of innovation on trait evolution may require synergism from other sources of ecological opportunity, like environmental/ecological setting[94]. Such synergisms may play out more widely across the vertebrate tree of life. For example, viviparous ray-finned fishes (Cyprinodontiformes), and cartilaginous fishes (Chondrichthyes), exhibit higher rates of diversification than their oviparous relatives, a product of their ability to colonize new ecological spaces[23,95]. Further, larger sizes in viviparous species than oviparous relatives are observed in other lineages, including spiny lizards, sharks, and rays[3,23]. Together, this study supports the notion that evolution of viviparity opens the opportunity to colonize new habitats and exploit available resources: these features together prompt the evolution of larger body sizes and/or faster rates of phenotypic diversification, shedding light on the mechanisms underpinning the high diversity of viviparous lineages in the most extreme thermal habitats in the world.

## Methods

### Phylogeny and species sampling
For all evolutionary analyses we used a previously published ultrametric tree of Liolaemidae[36]. That ultrametric tree was estimated using four mitochondrial and six nuclear loci, and contains 196 *Liolaemus* species, which represents ~70% of the lineage's recognized diversity. For each *Liolaemus* species included in the phylogeny, we gathered data for body size (snout-vent length; SVL), parity mode, substrate use, diet, and air environmental temperature (Supplementary Data 1) as we describe below.

### Body size (SVL)
*Liolaemus* species can exhibit sexual size dimorphism[83]; therefore, we focus this study only on the mean body size of females. To gather SVL information, we performed a search in Google Scholar with the following combinations: The scientific name for each species + body size, + snout vent length, or + SVL (for example, Liolaemus abaucan + body size). Because *Liolaemus* species inhabit many Latin American countries, and much of the corresponding literature is, therefore, in Spanish, we repeated the search as follows: The scientific name for each species + tamaño corporal, + longitud hocico cloaca, or + LHC. We used two criteria to choose mean body size of females from the literature to build our dataset. (1) We selected papers in which mean SVL from adult females and locality details were provided, allowing us to derive environmental layers from the same place in which lizard body size was measured. (2) When different papers reported mean SVL for the same species, we chose the paper with the highest sample size. The body size database was composed of 133 species, and for each of these species we gathered information of mean annual temperature, parity mode, substrate use, and diet, as we described below.

### Air temperature of environment
Body size of adult females was measured in one locality per species (Supplementary Data 1). For each of those localities we gathered data on general air temperature trends. Specifically, we extracted mean annual temperature (MAT; bio1) layer available in WorldClim dataset (resolution of 1 km$^2$)[96].

### Parity mode
We classified each species as either oviparous or viviparous based on a previous study[36]. In squamate reptiles, oviparity indicates that females lay eggs (i.e., egg-laying) and viviparity indicates that females give birth to hatchlings (i.e., live birth)[97].

### Substrate use
We categorized each species as arboreal, grass-bush dwelling, terrestrial (ground-dwelling and sand-dwelling), or saxicolous (rock- or boulder-dwelling) based on previous work[43]. For species in our SVL dataset, we determined that five species were arboreal, two were grass-bush dwelling species, 60 were saxicolous, and 63 were terrestrial. We were unable to determine substrate use for three species.

### Diet
We categorized each species as either insectivorous, omnivorous, or herbivorous, based on the most recently published database of diet in Liolaemidae[44]. In this database, each dietary category was determined based on the proportion of plant matter in the stomach content[44]. Specifically, species in the database were categorized as insectivorous when the plant consumption was lower than 11%, as omnivorous when the plant matter represents between 11 and 75% of the total diet, and as herbivorous when plants represent more than 75% of the diet[44]. For species not included in that data set, we performed a search in Google Scholar with the following combinations: The scientific name for each species + diet. We repeated the search in Spanish as follows: the scientific name for each species + dieta. For species in our SVL database, we determined that seven species were herbivorous, 64 were insectivorous, 48 were omnivorous, and we were unable to determine diet for 14 species.

### Evolutionary analyses
All evolutionary analyses (except the *MuSSCRat* analysis) were performed using the R environment for statistical computing, ver. 4.1.1[98].

**Multiple state-specific rates of continuous-character evolution.** We were interested in testing whether parity mode is associated with different rates of body size evolution. Therefore, we estimated the effect of parity mode (oviparity and viviparity) on rates of SVL evolution using a Bayesian, state-dependent, relaxed-clock model of Brownian motion (*MuSSCRat*)[99] implemented in RevBayes ver. 1.2.1[100]. This analysis allowed us to account for background rate variation across the tree, thereby reducing the risk of false positives, and jointly estimates the histories of discrete and continuous characters. The Markov chain Monte Carlo (MCMC) was run for 100 k generations with 10% burn-in, which we confirmed as sufficient to achieve effective sample size (ESS) > 200 for the model and key parameters. Since the model requires priors on the number of transitions in the discrete character (parity mode) and the number of rate shifts in the continuous character (body size), we performed analyses using different priors to evaluate their effect on the posterior estimates of key parameters. For both the discrete and the continuous character we used priors of 5, 15, and 25. As substrate use or diet (besides parity mode) also could influence rates of body size evolution, we performed two additional *MuSSCRat* analyses, in each using the same priors as above. In the first analysis, we compared rates of body size evolution between terrestrial and saxicolous species (excluding the arboreal and bush-dwelling species as they are very few in the database). In the second analysis, we compared rates of body size evolution between insectivorous and omnivorous/herbivorous species to test the hypothesis that the ingestion of plants (as occurs in both omnivorous and herbivorous species) influences body size evolution[33]. Therefore, in this and in the next evolutionary analyses (below) we combined omnivorous and herbivorous species. For the *MuSSCRat* and all evolutionary analyses, we used log$_{10}$ transformed SVL.

**Modeling stabilizing selection under Ornstein-Uhlenbeck models.** We were interested in testing whether parity mode is associated with different patterns of body size evolution. Therefore, we fitted a series

of Brownian motion (BM) and adaptive Ornstein-Uhlenbeck (OU) models to the body size data using the R package OUwie (ver. 2.6)[101] across 500 stochastic character maps of parity mode through the ultrametric tree using the *make.simmap* function in the R package phytools (ver. 1.0.3)[102]. In particular, we fitted five different evolutionary models to the trait data. BM1 is a single-rate model of stochastic trait evolution ($\sigma^2$) through the ultrametric tree, in which phenotypic differences among species evolves via a Brownian process, with no differences based on parity mode. BMS is a two-rate BM model in which the rate of stochastic character diffusion ($\sigma^2$) is allowed to vary between oviparous and viviparous species. The other three models were all adaptive Ornstein-Uhlenbeck evolution models. The simplest, OU1, is characterized by the presence of a single phenotypic optimum ($\theta$) for both parity modes and a shared rate of trait evolution ($\sigma^2$). OUM is also characterized by shared rate of evolution, but allows for the phenotypic optima to vary according to parity mode. The most complex model we fitted to the data, OUMV, allows for both the rate of trait evolution ($\sigma^2$) and the phenotypic optima ($\theta$) to vary between oviparous and viviparous species. The three OU models also include a single strength of convergence ($\alpha$) for oviparous and viviparous species. We compared the models through the Akaike information criterion corrected for small sample sizes (AICc)[103]. When two (or more) models were equally supported; $\Delta$AICc $\leq 2$, we focused on the results of the least-complex model. To account for phylogenetic uncertainty, we reran our OUwie analysis across 500 individually sampled trees from the posterior distribution (one stochastic character map of parity mode for each tree). As with parity mode, substrate use, or diet also could be associated with shifts in body size evolution. Following the same procedure as above, we also fitted the BM and OU models to substrate use (saxicolous and terrestrial categories) or diet (insectivorous and omnivorous/herbivorous categories). Lastly, we were interested in jointly comparing models according to parity mode, substrate use, and diet to explore which feature is a better predictor of body size shifts. Then, we reran the analyses with a subset of 111 species (with no missing data for parity mode, diet or substrate use) where BM and OU1 models were tested for all *Liolaemus* species, and BMS, OUM, and OUMV models were tested by parity mode, diet, and substrate use. In other words, we compared 11 different models as follows: BM, OU1, BMS (parity mode), BMS (substrate use), BMS (diet), OUM (parity mode), OUM (substrate use), OUM (diet), OUMV (parity mode), OUMV (substrate use), and OUMV (diet).

**Strength of convergent evolution.** In addition to testing whether an optimal body size predictably evolves with viviparity, we tested the strength (or lack thereof) of such convergence. To this end, we computed the Wheatsheaf index ($w$) using the *test.windex* function in the R package windex (ver. 2.0.3)[104]. The index measures the similarity of body size between viviparous species (focal group), where a high $w$ value indicates strong phenotypic convergence[105]. We assessed statistical significance of the test by running 500 bootstrap replicates.

**Phylogenetic path analysis.** We tested whether live birth is directly associated with body size shifts, or instead, whether body size shifts are associated with features correlated with viviparity. Studies in *Liolaemus* lizards have identified four relationships among viviparity, thermal habitat, substrate use, diet, and body size. These relationships are (1) viviparous species inhabit colder habitats than their oviparous counterparts[12,36,42], (2) whereas species from warm habitats tend to be terrestrial, species from cold habitats tend to thermoregulate on saxicolous substrates[43], (3) from an ancestral insectivorous diet in warm habitats, species from cool environments repeatedly evolved omnivorous/herbivorous diets (i.e., ingestion of plant matter)[33,44], and (4) species from cold habitats are larger than species from warm environments (i.e., Bergmann's rule)[56]. Based on these recognized

relationships, we proposed fifteen explanatory models about the direct, indirect, or direct/indirect effect of viviparity on body size of *Liolaemus* (Supplementary Fig. 2). Model 1: Viviparity is directly associated with body size in *Liolaemus*. Model 2 to 8: Viviparity is indirectly associated (mediated by environmental temperature) with body size in *Liolaemus*. In these models (except Model 4), environmental temperature also has an indirect effect (mediated by substrate use and/or diet) on body size. Model 9 to 15: Viviparity is directly and indirectly (mediated by environmental temperature) associated with body size in *Liolaemus*. In these models (except Model 11), environmental temperature also has a direct and indirect (mediated by substrate use and/or diet) effect on body size. Note that the direction in the association between viviparity and mean annual temperature in the models suggests that parity mode impacts distribution of species in different thermal habitats (viviparous species inhabit cold habitats), not that viviparity affects the environmental temperature itself. We simultaneously compared the fifteen proposed models by performing a phylogenetic path analysis implemented in the R package phylopath (ver. 1.1.3)[46]. This phylogenetic path analysis was performed with 111 species (~60% of the species included in the ultrametric tree of *Liolaemus*), for which we obtained data for all variables (body size, parity mode, air temperature of the environment, substrate use, and diet) used in this study. As in previous evolutionary analyses, we only included terrestrial and saxicolous species. Further, we categorized diet as insectivorous and omnivorous/herbivorous. We first compared the fifteen pre-defined models using Fisher's C statistic. The model is considered a good fit of the data if the C statistic is not significant ($p > 0.05$)[106]. The Fisher's C test[106] reflects the deviation of the data from the correlational structure predicted if the model is correct[45]. A two-sided $p < 0.05$ indicates that the data significantly deviates from the predicted model and, therefore, the model is not a good fit for the data. Based on this, different models have the potential to fit the same data, and model selection is then used to identify the best fit among the set of accepted path models[45]. We performed model selection by comparing the C-statistic information criterion for small sample sizes (CICc) and considered the best model the one with $\Delta$CICc $< 2$ and the lowest CICc[46,107].

**Stochastic linear Ornstein-Uhlenbeck models (SLOUCH).** Because we detected significant statistical associations between body size and environmental temperature (see results of Phylogenetic Path Analysis), we were interested in more deeply exploring the relationship between body size and the thermal habitat. To address this question, we simultaneously estimated an evolutionary regression, and an optimal regression in an OU framework using the *slouch.fit* function in the R package slouch (ver. 2.1.4)[67]. In this analysis, body size was the response trait, and air temperature of environment was the predictor trait. Our analyses estimated the phylogenetic half-life ($t_{1/2}$), the stationary variance ($V_y$), and the rate of adaptation ($\alpha$)[66,67]. A short ($t_{1/2}$), relative to the length of the ultrametric tree, and a higher $\alpha$ (far from 0, approaching or exceeding 100) represents instantaneous adaptation of body size to air temperature of environment[66]. By contrast, differences in the slope of the evolutionary and optimal regressions is supported when $t_{1/2}$ is bounded away from 0, indicating phylogenetic inertia, or a lag in adaptation of the trait to the predictor variable. We performed the SLOUCH analyses separately for viviparous and oviparous species to test whether phylogenetic half-life, stationary variance, and/or rate of adaptation differs between parity modes.

**Ancestral state reconstruction.** We performed ancestral state reconstruction using the *contMap* function in the R package phytools (ver. 1.0.3)[102] to graphically show evolution of body size. *Graphics*. Figures 1 and 2b were generated using the R package ggplot2

(ver. 3.3.6)[108] and edited using Adobe Illustrator. Figure 2a was generated using Adobe Illustrator.

## Reporting summary
Further information on research design is available in the Nature Portfolio Reporting Summary linked to this article.

## Data availability
Information of diet, mean body size of adult females, parity mode, and substrate use of *Liolaemus* lizards comes from previously published data. These details (including references of data sources), and mean annual air temperature are provided as Supplementary Data 1. Source data are provided with this paper.

## Code availability
The codes used to perform the evolutionary analyses are provided as Supplementary Code 1.

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

## Acknowledgements

This research was supported by project 61866 from the Templeton Foundation awarded to M.M.M.

## Author contributions

S.F.D.-G. and M.M.M. design the study. S.F.D.-G., D.E., and C.A.M.-M. build the database. S.F.D.-G., E.D.B., and L.R.V.A. performed the evolutionary analyses. S.F.D.-G. and M.M.M. drafted the manuscript. All authors contributed to subsequent revisions.

## Competing interests
The authors declare no competing interests.
