## [Peer Review File · Nature Communications]

Viviparity imparts a macroevolutionary signature of ecological opportunity in the body size of female *Liolaemus* lizardsReviewers' Comments:

Reviewer #1:

Remarks to the Author:

I read the manuscript entitled "Viviparity imparts a macroevolutionary signature of ecological opportunity in *Liolaemus* lizards" by Domínguez-Guerrero and colleagues. I found the manuscript interesting and well written. Concepts are clear and sound. However, I found several inaccuracies in the way the data were obtained or used; for example, it is said in the manuscript and therefore used in the analyses that *Liolaemus xanthoviridis* is a viviparous species, and it is an oviparous species. This error is also observed in Esquerré et al (2019). Other aspect that should be revised is body size, I understand that searching this information is difficult, but there are differences between the data the authors offered and other data in newer articles in the literature. For example, *L. abaucan* in Cabrera et al (2013) article in which I am a coauthor is 51.27 mm SVL in females, but several other papers consider this species above 60 mm; mean female SVL of *L. albiceps* in the description article (Lobo and Laurent, 1995) is 62.1 mm, but newer data for this species show 86.65 mm. I encourage the authors to search for more accurate data on SVL by diving in the supplementary data of some papers or asking collection managers for data. Then, diet data are treated in an awkward way, first the authors divide the species in three categories (insectivores, omnivores and herbivores), but in the analyses they used two categories (insectivores and omnivore-herbivores); additionally, there is not a clear explanation about how the authors determined each category. As a problematic example, it was shown in *L. crepuscularis* that diet may change in relation to availability, season or reproductive stage (Semhan et al., 2013). My suggestion is to determine a source of data (frequency, numerosity or volume), use a proportion or any other index and use this as a continuous variable (transformed if necessary) or set limits to determine each category. Finally, the authors used a single climatic variable (mean annual air temperature), why did the authors discarded other variables such as precipitation, elevation (in terms of oxygen availability)?; the authors used WorldClim data that offers more than mean annual air temperature.

On the conceptual aspect; there are at least five papers that deal with viviparity hypotheses in a phylogenetical context in *Liolaemidae* (Schulte et al., 2000; Medina and Ibargüengoytía, 2010; Pincheira Donoso et al., 2018; Esquerré et al., 2019 and Cruz et al., 2022), some of these papers were ignored, I strongly suggest to consider them. Thus, I recommend to revise the *L. xanthoviridis* reproductive mode, check as good as possible the SVL data of the studied species, determine on what are the dietary categories based and explain why the authors decided for only one climatic variable. As I said the manuscript is interesting, however the results obtained are not new; Pincheira Donoso et al. (2018) and Cruz et al. (2022) previously observed relationships with climate or other variables. Additionally, if the authors made a search for several aspects in *Liolaemus* species, I recommend to include reproductive mode, viviparity or other keywords related to the main topic of the manuscript in their search.

In summary, the concepts and the tools used to study how viviparous species colonized cold habitats in the case of *Liolaemus* species is interesting, but several crevices were noticed, as mentioned above. The source data should be revised, the novelty of the results should be considered and I would see some papers about the historical biogeography of *Liolaemidae* based on dispersal-vicariance analyses, to make sure what kind of climate might be the ancestral stage for this genus. I observed that the authors made a huge effort to collect information of the genus *Liolaemus*; however, some of this information lead to inaccuracies that ultimately may have a negative effect on the results (perhaps not). For this reason, I made a small search of literature in my folders that may be useful for the authors (I marked in with an asterisk those papers that may offer insights on the role of viviparity, some data on body size, diet and historical biogeography), which is listed at the end of my comments. If the authors cannot get these papers, they can contact me (Félix Cruz) and I can send a copy for it. Finally, I understand this is a preliminary version, but, please check the cited literature and set the scientific names in italics. I hope the authors find my comments as positive and encourage them to make their best effort to improve the quality of the data.

List of recommended literature

Aguilar C, Stark MR, Arroyo JA, Standing MD, Rios S, Washburn T, Sites JW Jr. Placental morphology in

two sympatric Andean lizards of the genus *Liolaemus* (Reptilia: Liolaemidae). *J Morphol.* 2015 Oct;276(10):1205-17. doi: 10.1002/jmor.20412. Epub 2015 Jul 29. PMID: 26220785.

Belver LC and Avila, LJ 2001 diet composition of *Liolaemus bibronii* (Iguania: Liolaemidae) in southern rio negro province, Argentina *Herpetological Journal*, Vol.11, Pp. 00-00 (2001)

Cabezas-Cartes F, Boretto JM, Ibargüengoytía NR. 2018 Effects of Climate and Latitude on Age at Maturity and Longevity of Lizards Studied by Skeletochronology. *Integr Comp Biol.* 2018 Dec 1;58(6):1086-1097. doi: 10.1093/icb/icy119. PMID: 30307522.

Cruz FB, DL Moreno Azócar, Vanhooydonck B, JA Schulte II, CS Abdala & A Herrel. 2021. Drivers and patterns of bite force evolution in Liolaemid lizards. *Biological Journal of the Linnean Society* 134: 126–140

*Cruz FB, Moreno Azócar DL, Perotti MG, Acosta JC, Stellatelli O, Vega L, Luna F, Antenucci D, Abdala C, Schulte II JA. 2022. The role of climate and maternal manipulation in determining and maintaining reproductive mode in *Liolaemus* lizards. *Journal of Zoology* 317: 101-113. doi.org/10.1111/jzo.12962

Cruz, FB; D Antenucci; F Luna; CS Abdala and LE Vega. 2011. Energetics in Liolaemini lizards: implications of a small body size and ecological conservatism. *Journal of Comparative Physiology B* 181:373–382.

*Díaz Gómez JM (2011) Estimating Ancestral Ranges: Testing Methods with a Clade of Neotropical Lizards (Iguania: Liolaemidae). *PLoS ONE* 6(10): e26412. doi:10.1371/journal.pone.0026412

Díaz Gómez, J.M. (2009), Historical biogeography of *Phymaturus* (Iguania: Liolaemidae) from Andean and Patagonian South America. *Zoologica Scripta*, 38: 1-7. <https://doi.org/10.1111/j.1463-6409.2008.00357.x>

*Díaz Gómez, Juan Manuel;; Lobo, Fernando; 2006 Historical Biogeography of a clade of *Liolaemus* (Iguania: Liolaemidae) based on ancestral areas and dispersal-vicariance analysis (DIVA). *PAPEIS AVULSOS DE ZOOLOGIA.*; Lugar: San Pablo, Brasil; Año: 2006 vol. 46 p. 261 - 274

Gómez Alés, R., Acosta, J.C., Astudillo, V. et al. 2018 Effect of temperature on the locomotor performance of species in a lizard assemblage in the Puna region of Argentina. *J Comp Physiol B* 188, 977–990 (2018). <https://doi.org/10.1007/s00360-018-1185-y>

Halloy, M.; Etheridge, R. & Burghardt, G. 1998. To bury in sand: phylogenetic relationships among lizard species of the Boulengeri group, *Liolaemus* (Reptilia: Squamata: Tropiduridae), based on behavioral characters. *Herpetological Monographs* 12: 1-37.

Jara, M., Frias-De-Diego, A., García-Roa, R. et al. The Macroecology of Chemical Communication in Lizards: Do Climatic Factors Drive the Evolution of Signalling Glands?. *Evol Biol* 45, 259–267 (2018). <https://doi.org/10.1007/s11692-018-9447-x>

Kozykariski M.L., L.C. Belver, L.J. Avila, Diet of the desert lizard *Liolaemus pseudoanomalus* (Iguania: Liolaemini) in northern La Rioja Province, Argentina, *Journal of Arid Environments*, Volume 75, Issue 11, 2011, Pages 1237-1239,

Labra A, Pienaar J, Hansen TF. 2009 Evolution of thermal physiology in *Liolaemus* lizards: adaptation, phylogenetic inertia, and niche tracking. *Am Nat.* 2009 Aug;174(2):204-20. doi: 10.1086/600088. PMID: 19538089.

*Medina M., N.R. Ibargüengoytía, 2010 How do viviparous and oviparous lizards reproduce in Patagonia? A comparative study of three species of *Liolaemus*, *Journal of Arid Environments*, Volume 74, Issue 9, 2010, Pages 1024-1032,

Moreno Azócar DL, Perotti MG, Bonino MF, Schulte II JA, Abdala CS, and Cruz FB. 2015. Body size and melanism: Compensatory traits? Variability in response to environmental factors in a lizards clade. *Journal of Zoology* 295: 243-253.

*O'Grady Shannon P., Mariana Morando, Luciano Avila, M. Denise Dearing, Correlating diet and digestive tract specialization: Examples from the lizard family Liolaemidae, *Zoology*, Volume 108, Issue 3, 2005, Pages 201-210,

Pincheira-Donoso, D. (2021). Correlated evolution between herbivory and gastrointestinal tract in a prolific lizard adaptive radiation. *Animal Biology*, 71(2), 233-241. <https://doi.org/10.1163/15707563-bja10051>

*Pincheira-Donoso, D. , S. F. Fox, J.A. Scolaro, N. Ibargüengoytía, J.C. Acosta, V. Corbalán, M. Medina, J. Boretto, H.J. Villavicencio & D.J. Hodgson. (2011). Body size dimensions in lizard ecological and evolutionary research: exploring the predictive power of mass estimation equations in two Liolaemidae

radiations. *Herpetological Journal*, 21, 35-42.

*Pincheira-Donoso, D., Jara, M., Reaney, A., García-Roa, R., Saldarriaga-Córdoba, M. & Hodgson, D.J. (2017). Hypoxia and hypothermia as rival agents of selection driving the evolution of viviparity in lizards. *Global Ecology and Biogeography*, 26, 1238-1246.

Pincheira-Donoso, D., Tregenza, T., Butlin, R.K. & Hodgson, D.J. (2018). Sexes and species as rival units of niche saturation during community assembly. *Global Ecology and Biogeography*, 27, 593-603.

*Reaney, A.M., Saldarriaga-Córdoba, M. & Pincheira-Donoso, D. (2018). Macroevolutionary diversification with limited niche disparity in a species-rich lineage of cold-climate lizards. *BMC Evolutionary Biology*, 18, 16.

Ruiz Monachesi, MR; Cruz, FB; Valdecantos S.; Labra, A. 2020. Unravelling associations among chemosensory system components in *Liolaemus* lizards. *Journal of Zoology* 312: 148-157

*Semhan Romina Valeria, Monique Halloy, and Cristian Simón Abdala "Diet and Reproductive States in a High Altitude Neotropical Lizard, *Liolaemus crepuscularis* (Iguania: Liolaemidae)," *South American Journal of Herpetology* 8(2), 102-108,

Tulli MJ, V Abdala and FB Cruz. 2011. Relationships among morphology, clinging performance and habitat use in *Liolaemini* lizards. *Journal of Evolutionary Biology* 24: 843-855

Tulli, M. J., F.B. Cruz, T. Kohlsdorf and V. Abdala. 2016. When a general morphology allows many habitat uses. *Integrative Zoology* 11: 473-489

Tulli, M.J. and F.B. Cruz. 2018. Is the number and size of scales in *Liolaemus* lizards driven by climate? *Integrative Zoology* 13: 579-594

Valdecantos María Soledad, Federico Arias, Robert E. Espinoza; Herbivory in *Liolaemus poecilochromus*, a Small, Cold-Climatic Lizard from the Andes of Argentina. *Copeia* 27 June 2012; 2012 (2): 203-210. doi: <https://doi.org/10.1643/CE-12-001>

Valdecantos S, V Martínez, F Lobo and F Cruz. 2013. Thermal biology of *Liolaemus* lizards from the high Andes: being efficient despite adversity. *Journal of Thermal Biology* 38:126-134.

*Valdecantos, M. S., F. Lobo, and V. Martínez. 2007. Estimación de edades, tamaño corporal y adquisición de la madurez sexual en dos especies de *Liolaemus* (Iguania: Liolaemidae). *Cuadernos de Herpetología* 21:31-44.

*Valdecantos, M.S., Lobo, F. (2007): Dimorfismo sexual en *Liolaemus multicolor* y *L. irregularis* (Iguania: Liolaemidae). *Rev. Esp. Herp.* 21: 55-69.

*Vanhooydonck B, FB Cruz, CS Abdala, DL Moreno Azócar, MF Bonino & A Herrel. 2010. Sex-specific evolution of bite performance in *Liolaemus* lizards (Iguania: Iguanidae): the battle of the sexes. *Biological Journal of the Linnean Society of London* 101: 461-475.

Villavicencio H.J, J.C. Acosta, M.A. Cánovas & J.A. Marinero (2003). Dimorfismo Sexual de *Liolaemus pseudoanomalus* (Iguania:Liolaemidae) en el Centro de Argentina. *Revista Española de Herpetología* 17: 87-92

Reviewer #2:

Remarks to the Author:

This paper addresses an interesting evolutionary question which is the association of reproductive strategy (oviparity or viviparity) with opportunities to colonize new habitats, and morphological evolution. The study uses a previously published and extensive phylogeny of *Liolaemus* lizards as the basis for this work. *Liolaemus* lizards are an appropriate place to test for these associations. The group is speciose, with wide distribution and variation in reproductive strategy as well as habitat and body size. The specific question being addressed is whether body size evolution is related to reproductive strategy, or whether other attributes correlated with viviparity are associated with body size evolution. The results show that viviparity is associated with faster body size evolution than oviparity, but that the effects of viviparity on body size evolution are indirect (via changes in thermal regime and habitat use).

The paper is concise and well written, and the figures and tables are useful. The work will be of significance to the field because we do not yet completely understand the evolutionary history and consequences of viviparity. Viviparity is a significant adaptation because it has evolved so many times.

Incidentally, the ~120 origins of viviparity to which this paper refers should be corrected to in "reptiles", or alternatively corrected to ~115 origins in squamate. There are 6 origins in extinct reptile lineages.

The paper uses a phylogenetic modelling approach. The three models tested examined direct/indirect/both influences of viviparity on body size. One question is whether the other direction of influence should be examined. Could not body size evolution have influences on viviparity evolution? It would be straightforward to test for this association using this dataset to settle a 'chicken and egg' problem. Larger body size might compensate for inability to produce multiple clutches per season (possible in oviparity) by higher fecundity thus facilitating extended egg retention.

Other comments

L36: viviparous females can also influence offspring sex Robert, K. A. and M. B. Thompson (2001).

"Sex determination: Viviparous lizard selects sex of embryos." *Nature* 412(6848): 698-699. Wapstra,

E., et al. (2004). "Maternal basking behaviour determines offspring sex in a viviparous reptile."

Proceedings of the Royal Society of London. Series B: Biological Sciences 271(suppl_4): S230-S232.

44: reword 'simplified' for clarity

60: conflates chondrichthyans with squamates; specify "in chondrichthyans" at reference 21 and "in squamates" at 22/23

95: "optimal body size" needs to be defined.

101: How might structural resources constitute ecological opportunity in cartilaginous and ray-finned fish? Their habitat use is rather different to squamates.

Methods: the systematic literature searches could be formatted in quotation marks for clarity.

Parity mode: Could the authors elaborate on classification of reproductive strategy from the previous study. There are standardized criteria: Blackburn, D. G. (1993). "Standardized criteria for the recognition of reproductive modes in squamate reptiles." *Herpetologica* 49(1): 118-132.

Air temperature: the study species utilize basking for temperature regulation. Therefore, would it be appropriate to also include hours of sunlight (that is, potential solar exposure) in the model.

134: the systematic nature of the study requires more information about the source of species' descriptions

What is the effect of missing data on the model?

148: which priors were used for repeated analyses of the number of character transitions

198: could the percentage of total *Liolaemus* species be provided?

Figure 2 legend explain boxplot parameters

Reviewer #3:

Remarks to the Author:

The authors present a rich dataset including 196 species of *Liolaemus* lizards, their body size, substrate use, diet, and parity mode to test whether parity mode affects body size evolution and diversification. They wonderfully integrate these data with sophisticated, cutting edge, and appropriate phylogenetic comparative methods, and natural history knowledge. They show that viviparity has allowed the lizards to colonize cold habitat niches, although has only modestly affected body size evolution. Although connections between body size, parity mode, diet, and cold environments have long been considered and studied, the authors bring a new perspective through integrating all of these factors in their path analyses, along with possibly the most extensive dataset. This work is certainly worth publishing and makes a great contribution.

I have two general comments that the authors should consider, followed by line-specific feedback. The general comments are:

1. There is a slight mismatch in the message in the results and the conclusion. The message in the results seemed to be that viviparity does affect body size evolution, which I had hesitations about. Then the last paragraphs indicates that the effect is not very strong, which I agree with, but seems to be different from how the results are written. I think this is mostly an issue with messaging in the

results. The path analysis shows only an indirect effect of parity on body size (really none of the variables have strong effects on body size). The OU modeling shows that the OU1 model, which is really a null model, is not much worse than the OUM and OUMV model. This comment can be addressed simply by making the message that body size is only weakly related to other variables more clear in the results section.

2. Including a wider range of models in the BM/OU modeling and possibly the path analysis may provide more insights around what shapes body size evolution.

a. Currently, the OU modeling includes BM1, OU1, BMS, OUM, OUMV. First two are null, last three are centered on parity mode. I suggest a wider set of models that consider alternative hypotheses for which the authors have data and that they talk about in the discussion. In particular, other adaptive models that might allow the authors to tease apart what is going on are BMS, OUM, and OUMV using diet and substrate (separately). For example, it is possible that an OUM model with diet ends up being the best, and this would suggest that body size evolution is driven by diet and not parity mode. The fact is that currently only a small set of models are considered and one of the null models (OU1) is not much worse than the adaptive models for parity mode. No really viable alternative hypothesis is considered. A possible non-adaptive model that the authors could consider is BMS, OUM, OUMV with different clades identified. The branch colors in Figure 1 suggest that there might be more of clade-specific shifts in body size evolutionary rates.

b. The three models considered in the path analysis are all very similar and great at teasing out how parity might or might not affect body size. Some additional models could be used to understand the effects of the other variables already included in the analysis. One thing that I find striking in the path analysis (Figure 3) is that none of the factors seem to have a very strong effect on body size evolution (only mean annual temperature has a significant effect, but it does not seem very strong).

Specific comments:

Abstract:

- On line 17, the authors mention that ecological opportunity may "reconfigure phenotypic diversity", but this idea does not seem to be revisited in the abstract, at least not explicitly. I suspect that this is connected to larger body size in viviparous lizards, which seems less a reconfiguration. I would make that connection more explicit.

Introduction:

Line 39: Delete "colonize", as access than colonize give the same meaning here.

Lines 39-42: This sentence unrealistically simplifies the situation: there certainly are temperate oviparous species as well as plenty of viviparous tropical species of squamates.

Line 87: "repeated" (delete "ly").

Line 93: "another" should be "other".

Material and Methods:

Lines 120-123: It sounds like air temperature was sampled such that there was one number per species based on one individual from which SVL was collected. I would clarify that this is the case. One could imagine other approaches, like identifying a geographic range based on museum specimens available and taking the average air temperature across all pixels within the range.

Line 145: Was a 10% burn-in adequate? Was this evaluated in some way?

Line 147: The authors mention key parameters (plural), but then list only a single parameter, the number of rate shifts, for which they tried three different values. I have a couple questions about this. First, were there any other parameters for which priors were required? If so, what were they and how were they treated? Second, Bayesian approaches typically view parameters as distributions, so it is strange to provide a single value (like 5, 10 or 20) to characterize a prior. Was there a distribution to the prior? What were the characteristics of that distribution?

Lines 159-164: OU models also have a parameter, alpha, which is the strength of selection to an optimum, but the authors do not mention it. This should be included. Although OUwie also allows for fitting of OUMA models that allow alpha to differ between parity modes, I do not think that is

necessary here.

Lines 204-206: The use of Fischer's C statistic and corresponding p-value is unclear. The authors state that a low p-value indicates that the available evidence rejects the model. What is the null hypothesis of this test? What is it evaluating? Is this a model fit test of some sort?

Results:

Line 230: What the posterior probability represents is unclear. Looking at figure 1, I assume that the boxplots in the top left represent the posterior probability distributions for the rate parameter for oviparous and viviparous species. Does the posterior probability (0.77-0.89) represent the probability that the two rates are different? Whether this is the case or not, this needs to be explained better.

Line 230: I suggest staying in past tense, so "find" should be "found" (and other places).

Lines 231-233: The headings to supplementary tables 1 and 2 look the same. I suggest modifying the beginning to explain what alternative phylogeny is being used in the second table. Also, looking at these tables (especially supplementary table 1), it is worth considering that the 3rd best model, which suggests no difference in SVL between oviparous and viviparous species, is also not far behind, with a delta of three. This suggests that the support for different body size between the two groups is moderate. Is modeling averaging possible here? It was done with the path analysis?

Lines 245-246: I suggest more clearly explaining what is meant by the optimal and evolutionary regressions being decoupled. Looking at supplementary table 4 it looks like none of the slopes are significantly different from zero (the SE of the slopes are only slightly lower than the slopes themselves).

Discussion:

Line 286: Including when the Andes began forming would provide useful context for this discussion of dates. Did the Andes predate, coincide with, or postdate the origin of *Liomaemus*?

Line 296: Although temperature is a direct predictor of body size and is the only one in the path model, the effect seems somewhat weak (Fig. 3).

Line 313: Larger species also often feed on a larger range of prey sizes.

Lines 314-315: It seems that the authors have the data to test some of this, for example, an OU model for body size evolution that has different optima for different diets.

Lines 359-360: I suggest rephrasing this statement to communicate right away that Bergmann's rule is supported in only some groups, and the results from reptiles are very much mixed. I know this appears a few lines later, but the statement in these lines seems too bold.

Lines 392-393: I agree with this statement, but feel as though the results of the paper, while accurately describing the results were trying to convince me that viviparity does influence body size evolution (for example, supplementary table 1, in main figure 2).

Line 393: Should it read "...influence body size evolution directly...?"

Figure 1: This is a great figure, but some basic modifications would make it easier to understand and interpret. The caption should indicate that the boxplots are posterior probability distributions of the Brownian rate parameter (σ), that the branches are colored by rates of evolution, and explain the temperature bar along with the lines connecting to it from the species data. I also suggest that the black scale bar for SVL be broken down a little with some visual guide going across the species data. For example, use two different shades of grey to show 0, 50, and 100 mm points and then use a transparent filled box or a light line (maybe dashed) going down from each point so that it is easier to see how big the SVL of species further down is. Finally, the colors used for omnivore and herbivore are a little difficult to distinguish.

Figure 2: This is another great figure, but the 500 points blend in to form green and purple blobs. I suggest making the points smaller and seeing if there are other ways to make them more distinct.

Supplementary Table 3: The meaning of the c-statistic and associated p-value should be mentioned here as well.

Synopsis: We want to thank the three reviewers for their clear and helpful feedback and in this new version we addressed point-by-point all the reviewers' comments. The two major points raised by the reviewers were 1) that the dataset required revision and 2) that additional evolutionary models should be considered. We appreciate these concerns and we carefully reviewed our dataset to verify parity mode, substrate use and diet of each species. Further, for each species we did a new search of mean SVL for adult females. As a result, we modified SVL information for some species when data were supported by larger sample sizes, or we deleted some SVL data that did not explicitly correspond with average body size of adult females. Then, we reran all our analyses, and we included additional models as suggested by the reviewers. We believe these revisions helped us produce a stronger manuscript, and we appreciated the constructive feedback.

REVIEWER COMMENTS

Reviewer #1 (Remarks to the Author):

Comment:

I read the manuscript entitled “Viviparity imparts a macroevolutionary signature of ecological opportunity in *Liolaemus* lizards” by Domínguez-Guerrero and colleagues. I found the manuscript interesting and well written. Concepts are clear and sound. However, I found several inaccuracies in the way the data were obtained or used; for example, it is said in the manuscript and therefore used in the analyses that *Liolaemus xanthoviridis* is a viviparous species, and it is an oviparous species. This error is also observed in Esquerré et al (2019).

Response:

Thank you for this observation. In our revision, we coded *L. xanthoviridis* as oviparous. We also revised parity mode for the rest of the species included in our dataset, and our parity mode data correctly align with other databases¹⁻³.

Comment:

Other aspect that should be revised is body size, I understand that searching this information is difficult, but there are differences between the data the authors offered and other data in newer articles in the literature.

Response:

We appreciate the concern of the reviewer that our estimates of body size for some species in our dataset (which come from published papers) differ from reported body size for the same species in other published studies. After evaluating the works listed by the reviewer and other studies on these lizards, we suspect the discrepancies the reviewer noted may have arisen because we focused only on the average SVL of adult females whereas other works provided maximal SVL, estimated the mean SVL using the largest two-thirds of adult females, or combined SVL data for both males and females. Other differences in body size among studies might be also related to intraspecific variation among populations. In the revision, we performed a new literature search

of mean SVL data of females (including the list of papers that the reviewer supplied) of each *Liolaemus* species to ensure we had the most up-to-date and accurate dataset for average adult female body size. We used two criteria (now detailed in the Methods section) to choose mean SVL of females from the literature to build our dataset. 1) We selected papers in which mean SVL from adult females and locality details were provided, allowing us to derive environmental layers from the same place in which lizard body size was measured. 2) When different papers reported mean SVL for the same species, we chose the paper with the highest sample size.

Comment:

For example, *L. abaucan* in Cabrera et al (2013) article in which I am a coauthor is 51.27 mm SVL in females, but several other papers consider this species above 60 mm

Response:

Thank you for the observation. The SVL of *L. abaucan* reported in Cabrera et al. (2013)⁴ was estimated using the larger third of the individuals. We searched for additional SVL data for *L. abaucan* and we found SVL data in other six studies^{3,5-9}. Three of these studies reported SVL equal to or higher than 60mm^{5,6,9}, and the other three reported SVL lower than 60 mm^{3,7,8}. However, five of those six studies provided information about maximal SVL in females⁵, maximal SVL for the species^{6,8}, or average SVL for males and females combined^{7,9}. Only one study provided data about mean SVL of adult females (53.5 mm)³, which we used as the source of body size data for *L. abaucan* in our dataset.

Comment:

Mean female SVL of *L. albiceps* in the description article (Lobo and Laurent, 1995) is 62.1 mm, but newer data for this species show 86.65 mm.

Response:

We found SVL data for *L. albiceps* in nine studies (in addition to Lobo & Laurent, 1995)¹⁰ and, as the reviewer notes, some of those studies reported body size to be greater than 80 mm for this species^{4,6,7,9,11-15}. However, those data reflect mean SVL for males¹⁴, mean SVL for both sexes^{7,9,15}, maximal SVL for *L. albiceps*^{6,11,12}, or did not specify whether the data were from males, females, or both^{13,16}. Only one of these nine studies provided mean SVL for females (69.52, n=7)⁴, although this study did not provide coordinates or information about the sampling locality from which the body size data were gathered. By contrast, the work of Lobo and Laurent (1995)¹⁰ provided information about the locality from which the body size were gathered, and a relatively large sample size for SVL data (n= 35 females), so we considered this study the most appropriate source of SVL data for *L. albiceps*.

Comment:

I encourage the authors to search for more accurate data on SVL by diving in the supplementary data of some papers or asking collection managers for data.

Response:

We appreciate the reviewer's helpful suggestion. We performed a new search for mean SVL data of females for each *Liolaemus* species, which allowed us to improve our database in the revision.

Comment:

Then, diet data are treated in an awkward way, first the authors divide the species in three categories (insectivores, omnivores and herbivores), but in the analyses they used two categories (insectivores and omnivore-herbivores); additionally, there is not a clear explanation about how the authors determined each category.

Response:

We used the most recently published database of diet in Liolaemidae¹⁷, which broadly aligns with previous diet databases^{18,19}. From this database, we categorized each species as either insectivorous, omnivorous or herbivorous. For the evolutionary analyses, we only included two categories (insectivorous and omnivorous-herbivorous) because in lizards there is a hypothesis that the ingestion of plants (as occurs in both omnivorous and herbivorous species) influences the evolution of body size¹⁸. So, in our coding scheme, the inclusion of plants into the diet was the differentiating factor among dietary categories. Whereas omnivorous and insectivorous species represent 43% and 52% of the species included in our dataset, herbivorous species represent only 5% of *Liolaemus* species. Given the low number of herbivorous species in our dataset and analysis, this should not have an important effect on our results. To check this, we reran our phylogenetic path analysis including only insectivorous and omnivorous species and we found a consistent pattern: larger sizes in viviparous species are associated with low environmental temperatures, but they are not associated with diet, substrate use, or parity mode. We describe how dietary classifications from the published databases were determined in response to a similar question below.

Comment:

As a problematic example, it was shown in *L. crepuscularis* that diet may change in relation to availability, season or reproductive stage (Semhan et al., 2013). My suggestion is to determine a source of data (frequency, numerosity or volume), use a proportion or any other index and use this as a continuous variable (transformed if necessary) or set limits to determine each category.

Response:

We agree that focusing on proportion diet data is important to determinate the dietary categories. For this reason, we based our diet dataset from a recently published database in which each dietary category was determined based on the proportion of plant matter in the stomach content¹⁷. In short, species were categorized as insectivorous when the plant consumption was lower than 11%, as omnivorous when the plant matter represents between 11 and 75%, and as herbivorous when plants represent more than 75% of the diet¹⁷. We provide these details in the Methods section of our revision so that readers need not track down the data from the original source, and apologize for the confusion this may have caused in our first submission.

Comment:

Finally, the authors used a single climatic variable (mean annual air temperature), why did the authors discarded other variables such as precipitation, elevation (in terms of oxygen availability)?; the authors used WorldClim data that offers more than mean annual air temperature.

Response:

Environmental temperature is the most recognized driver of live birth in lizards²⁰⁻²² and simultaneously is associated with body size in animals²³⁻²⁶. Therefore, mean annual air temperature is an environmental variable that allows us to bridge between viviparity and body size within our phylogenetic path analysis. To explore whether precipitation and/or elevation connect parity mode and body size we reran our phylogenetic path analysis including precipitation or elevation instead of mean annual air temperature. When we included precipitation or elevation, all models were statistically significant (p -values <0.05), meaning they are do not fit the data well. Therefore, we only included mean annual air temperature within our phylogenetic path analysis.

Comment:

On the conceptual aspect; there are at least five papers that deal with viviparity hypotheses in a phylogenetical context in Liolaemidae (Schulte et al., 2000; Medina and Ibargüengoytía, 2010; Pincheira Donoso et al., 2018; Esquerré et al., 2019 and Cruz et al., 2022), some of these papers were ignored, I strongly suggest to consider them.

Response:

We agree these should be in the paper. Two of these were already cited in our original submission, and we included citations for the other three papers in the revision.

Comment:

Thus, I recommend to revise the *L. xanthoviridis* reproductive mode, check as good as possible the SVL data of the studied species, determine on what are the dietary categories based and explain why the authors decided for only one climatic variable.

Response:

See responses above.

Comment:

As I said the manuscript is interesting, however the results obtained are not new; Pincheira Donoso et al. (2018) and Cruz et al. (2022) previously observed relationships with climate or other variables.

Response:

Certainly, these relevant papers and several other supports relationships between body size and/or parity mode with climate. We believe our studies builds on these strong previous studies by disentangling direct and indirect effects using an evolutionary causal approach. Through this approach, we infer that larger body sizes in viviparous species are product of their overrepresentation in cool habitats and not owing to the evolution of live birth per se. In other

words, we discovered that the ecological opportunity furnished by the evolution of live birth (colonization of cold habitats) produces predictable phenotypic shifts.

Comment:

Additionally, if the authors made a search for several aspects in *Liolaemus* species, I recommend to include reproductive mode, viviparity or other keywords related to the main topic of the manuscript in their search.

Response:

We incorporated this suggestion in our new search.

Comment:

In summary, the concepts and the tools used to study how viviparous species colonized cold habitats in the case of *Liolaemus* species is interesting, but several crevices were noticed, as mentioned above. The source data should be revised, the novelty of the results should be considered and I would see some papers about the historical biogeography of *Liolaemidae* based on dispersal-vicariance analyses, to make sure what kind of climate might be the ancestral stage for this genus. I observed that the authors made a huge effort to collect information of the genus *Liolaemus*; however, some of this information lead to inaccuracies that ultimately may have a negative effect on the results (perhaps not). For this reason, I made a small search of literature in my folders that may be useful for the authors (I marked in with an asterisk those papers that may offer insights on the role of viviparity, some data on body size, diet and historical biogeography), which is listed at the end of my comments. If the authors cannot get these papers, they can contact me (Félix Cruz) and I can send a copy for it. Finally, I understand this is a preliminary version, but, please check the cited literature and set the scientific names in italics. I hope the authors find my comments as positive and encourage them to make their best effort to improve the quality of the data.

Response:

We did indeed find the comments positive and encouraging. We are very grateful for all the suggestions and help with additional body size data that allowed us to improve the paper.

List of recommended literature

Aguilar C, Stark MR, Arroyo JA, Standing MD, Rios S, Washburn T, Sites JW Jr. Placental morphology in two sympatric Andean lizards of the genus *Liolaemus* (Reptilia: *Liolaemidae*). *J Morphol.* 2015 Oct;276(10):1205-17. doi: 10.1002/jmor.20412. Epub 2015 Jul 29. PMID: 26220785.

Belver1 LC and Avila, LJ 2001 diet composition of *liolaemus bibronii* (iguania: *liolaemidae*) in southern rio negro province, argentina *herpetological journal*, Vol.11, Pp. 00-00 (2001)

Cabezas-Cartes F, Boretto JM, Ibargüengoytía NR. 2018 Effects of Climate and Latitude on Age at Maturity and Longevity of Lizards Studied by Skeletochronology. *Integr Comp Biol.* 2018 Dec 1;58(6):1086-1097. doi: 10.1093/icb/icy119. PMID: 30307522.

Cruz FB, DL Moreno Azócar, Vanhooydonck B, JA Schulte II, CS Abdala & A Herrel. 2021. Drivers and patterns of bite force evolution in *Liolaemid* lizards. *Biological Journal of the Linnean Society* 134: 126–140

***Cruz FB, Moreno Azócar DL, Perotti MG, Acosta JC, Stelatelli O, Vega L, Luna F,**

Antenucci D, Abdala C, Schulte II JA. 2022. The role of climate and maternal manipulation in determining and maintaining reproductive mode in Liolaemus lizards. *Journal of Zoology* 317: 101-113. doi.org/10.1111/jzo.12962

Cruz, FB; D Antenucci; F Luna; CS Abdala and LE Vega. 2011. Energetics in Liolaemini lizards: implications of a small body size and ecological conservatism. *Journal of Comparative Physiology B* 181:373–382.

***Díaz Gómez JM (2011) Estimating Ancestral Ranges: Testing Methods with a Clade of Neotropical Lizards (Iguania: Liolaemidae). PLoS ONE 6(10): e26412. doi:10.1371/journal.pone.0026412**

Díaz Gómez, J.M. (2009), Historical biogeography of Phymaturus (Iguania: Liolaemidae) from Andean and patagonian South America. *Zoologica Scripta*, 38: 1-7. <https://doi.org/10.1111/j.1463-6409.2008.00357.x>

***Díaz Gómez, Juan Manuel;; Lobo, Fernando; 2006 Historical Biogeography of a clade of Liolaemus (Iguania: Liolaemidae) based on ancestral areas and dispersal-vicariance analysis (DIVA). PAPEIS AVULSOS DE ZOOLOGIA.; Lugar: San Pablo, Brasil; Año: 2006 vol. 46 p. 261 - 274**

Gómez Alés, R., Acosta, J.C., Astudillo, V. et al. 2018 Effect of temperature on the locomotor performance of species in a lizard assemblage in the Puna region of Argentina. *J Comp Physiol B* 188, 977–990 (2018). <https://doi.org/10.1007/s00360-018-1185-y>

Halloy, M.; Etheridge, R. & Burghardt, G. 1998. To bury insand: phylogenetic relationships among lizard species of the boulengeri group, Liolaemus (Reptilia: Squamata: Tropiduridae), based on behavioral characters. *Herpetological Monographs* 12: 1-37.

Jara, M., Frias-De-Diego, A., García-Roa, R. et al. The Macroecology of Chemical Communication in Lizards: Do Climatic Factors Drive the Evolution of Signalling Glands?. *Evol Biol* 45, 259–267 (2018). <https://doi.org/10.1007/s11692-018-9447-x>

Kozykariski M.L., L.C. Belver, L.J. Avila, Diet of the desert lizard Liolaemus pseudoanomalus (Iguania: Liolaemini) in northern La Rioja Province, Argentina, *Journal of Arid Environments*, Volume 75, Issue 11, 2011, Pages 1237-1239,

Labra A, Pienaar J, Hansen TF. 2009 Evolution of thermal physiology in Liolaemus lizards: adaptation, phylogenetic inertia, and niche tracking. *Am Nat.* 2009 Aug;174(2):204-20. doi: 10.1086/600088. PMID: 19538089.

***Medina M., N.R. Ibargüengoytía, 2010 How do viviparous and oviparous lizards reproduce in Patagonia? A comparative study of three species of Liolaemus, Journal of Arid Environments, Volume 74, Issue 9,2010, Pages 1024-1032,**

Moreno Azócar DL, Perotti MG, Bonino MF, Schulte II JA, Abdala CS, and Cruz FB. 2015. Body size and melanism: Compensatory traits? Variability in response to environmental factors in a lizards clade. *Journal of Zoology* 295: 243-253.

***O'Grady Shannon P., Mariana Morando, Luciano Avila, M. Denise Dearing, Correlating diet and digestive tract specialization: Examples from the lizard family Liolaemidae, Zoology, Volume 108, Issue 3, 2005, Pages 201-210,**

Pincheira-Donoso, D. (2021). Correlated evolution between herbivory and gastrointestinal tract in a prolific lizard adaptive radiation. *Animal Biology*, 71(2), 233-241. <https://doi.org/10.1163/15707563-bja10051>

- *Pincheira-Donoso, D. , S. F. Fox, J.A. Scolaro, N. Ibarzüengoytía, J.C. Acosta, V. Corbalán, M. Medina, J. Boretto, H.J. Villavicencio & D.J. Hodgson. (2011). Body size dimensions in lizard ecological and evolutionary research: exploring the predictive power of mass estimation equations in two Liolaemidae radiations. *Herpetological Journal*, 21, 35-42.**
- *Pincheira-Donoso, D., Jara, M., Reaney, A., García-Roa, R., Saldarriaga-Córdoba, M. & Hodgson, D.J. (2017). Hypoxia and hypothermia as rival agents of selection driving the evolution of viviparity in lizards. *Global Ecology and Biogeography*, 26, 1238-1246.**
- Pincheira-Donoso, D., Tregenza, T., Butlin, R.K. & Hodgson, D.J. (2018). Sexes and species as rival units of niche saturation during community assembly. *Global Ecology and Biogeography*, 27, 593-603.**
- *Reaney, A.M., Saldarriaga-Córdoba, M. & Pincheira-Donoso, D. (2018). Macroevolutionary diversification with limited niche disparity in a species-rich lineage of cold-climate lizards. *BMC Evolutionary Biology*, 18, 16.**
- Ruiz Monachesi, MR; Cruz, FB; Valdecantos S.; Labra, A. 2020. Unravelling associations among chemosensory system components in Liolaemus lizards. *Journal of Zoology* 312: 148–157**
- *Semhan Romina Valeria, Monique Halloy, and Cristian Simón Abdala "Diet and Reproductive States in a High Altitude Neotropical Lizard, *Liolaemus crepuscularis* (Iguania: Liolaemidae)," *South American Journal of Herpetology* 8(2), 102-108,**
- Tulli MJ, V Abdala and FB Cruz. 2011. Relationships among morphology, clinging performance and habitat use in Liolaemini lizards. *Journal of Evolutionary Biology* 24: 843-855
- Tulli, M. J., F.B. Cruz, T. Kohlsdorf and V. Abdala. 2016. When a general morphology allows many habitat uses. *Integrative Zoology* 11: 473-489
- Tulli, M.J. and F.B. Cruz. 2018. Is the number and size of scales in Liolaemus lizards driven by climate? *Integrative Zoology* 13: 579–594
- Valdecantos María Soledad, Federico Arias, Robert E. Espinoza; Herbivory in *Liolaemus poecilochromus*, a Small, Cold-Climatic Lizard from the Andes of Argentina. *Copeia* 27 June 2012; 2012 (2): 203–210. doi: <https://doi.org/10.1643/CE-12-001>
- Valdecantos S, V Martínez, F Lobo and F Cruz. 2013. Thermal biology of *Liolaemus* lizards from the high Andes: being efficient despite adversity. *Journal of Thermal Biology* 38:126-134.
- *Valdecantos, M. S., F. Lobo, and V. Martí'nez. 2007. Estimación de edades, tamaño corporal y adquisición de la madurez sexual en dos especies de *Liolaemus* (Iguania: Liolaemidae). *Cuadernos de Herpetología* 21:31–44.**
- *Valdecantos, M.S., Lobo, F. (2007): Dimorfismo sexual en *Liolaemus multicolor* y *L. irregularis* (Iguania: Liolaemidae). *Rev. Esp. Herp.* 21: 55-69.**
- *Vanhooydonck B, FB Cruz, CS Abdala, DL Moreno Azócar, MF Bonino & A Herrel. 2010. Sex-specific evolution of bite performance in *Liolaemus* lizards (Iguania: Iguanidae): the battle of the sexes. *Biological Journal of the Linnean Society of London* 101: 461–475.**
- Villavicencio H.J, J.C. Acosta, M.A. Cánovas & J.A. Marinero (2003). Dimorfismo Sexual de *Liolaemus pseudoanomalus* (Iguania:Liolaemidae) en el Centro de Argentina. *Revista Española de Herpetología* 17: 87-92

Reviewer #2 (Remarks to the Author):

Comment:

This paper addresses an interesting evolutionary question which is the association of reproductive strategy (oviparity or viviparity) with opportunities to colonize new habitats, and morphological evolution. The study uses a previously published and extensive phylogeny of *Liolaemus* lizards as the basis for this work. *Liolaemus* lizards are an appropriate place to test for these associations. The group is speciose, with wide distribution and variation in reproductive strategy as well as habitat and body size. The specific question being addressed is whether body size evolution is related to reproductive strategy, or whether other attributes correlated with viviparity are associated with body size evolution. The results show that viviparity is associated with faster body size evolution than oviparity, but that the effects of viviparity on body size evolution are indirect (via changes in thermal regime and habitat use).

The paper is concise and well written, and the figures and tables are useful. The work will be of significance to the field because we do not yet completely understand the evolutionary history and consequences of viviparity.

Response:

We appreciate the Reviewer's feedback and kind words.

Comment:

Viviparity is a significant adaptation because it has evolved so many times. Incidentally, the ~120 origins of viviparity to which this paper refers should be corrected to in "reptiles", or alternatively corrected to ~115 origins in squamate. There are 6 origins in extinct reptile lineages.

Response:

Corrected. Thank you for the observation.

Comment:

The paper uses a phylogenetic modelling approach. The three models tested examined direct/indirect/both influences of viviparity on body size. One question is whether the other direction of influence should be examined. Could not body size evolution have influences on viviparity evolution? It would be straightforward to test for this association using this dataset to settle a 'chicken and egg' problem. Larger body size might compensate for inability to produce multiple clutches per season (possible in oviparity) by higher fecundity thus facilitating extended egg retention.

Response:

This is a good point, as larger sizes may be associated with evolution of live birth, for example, in sharks²⁷. In the revision, we included all the potential paths linking viviparity directly or indirectly with SVL, resulting in 15 models (Model 1-15; Figure 1). Additionally, we included 7 models in which SVL could influence parity mode (Model 16-22; Figure 1). We ran the phylogenetic path analysis and none of the models including SVL as potential driver of parity

mode were statistically supported (p -values < 0.001). Therefore, we only included the first 15 models in our manuscript.

Figure 1. Models describing the direct, indirect, and direct/indirect effects of viviparity on body size (Models 1-15), or the potential effect of body size on parity mode transitions (Models 16-22) in *Liolaemus* lizards.

Comment:

Other comments

L36: viviparous females can also influence offspring sex Robert, K. A. and M. B. Thompson (2001). "Sex determination: Viviparous lizard selects sex of embryos." *Nature* 412(6848): 698-699. Wapstra, E., et al. (2004). "Maternal basking behaviour determines offspring sex in a viviparous reptile." *Proceedings of the Royal Society of London. Series B: Biological Sciences* 271(suppl_4): S230-S232.

Response:

Thank you for these suggestions. We have added them to the paper.

Comment:

44: reword 'simplified' for clarity

Response:

Done.

Comment:

60: conflates chondrichthyans with squamates; specify “in chondrichthyans” at reference 21 and “in squamates” at 22/23

Response:

Specified.

Comment:

95: “optimal body size” needs to be defined.

Response:

Done.

Comment:

101: How might structural resources constitute ecological opportunity in cartilaginous and ray-finned fish? Their habitat use is rather different to squamates.

Response:

We modified that sentence for clarity.

Comment:

Methods: the systematic literature searches could be formatted in quotation marks for clarity.

Response:

Done.

Comment:

Parity mode: Could the authors elaborate on classification of reproductive strategy from the previous study. There are standardized criteria: Blackburn, D. G. (1993). "Standardized criteria for the recognition of reproductive modes in squamate reptiles." *Herpetologica* 49(1): 118-132.

Response:

In the revision we included the criteria to distinguish oviparous and viviparous species based on the recommended reference.

Comment:

Air temperature: the study species utilize basking for temperature regulation. Therefore, would it be appropriate to also include hours of sunlight (that is, potential solar exposure) in the model.

Response:

This is a good suggestion as viviparous species in *Liolaemus* exhibit greater activity time than oviparous species². To explore whether activity hours connect parity mode and body size we reran our phylogenetic path analysis including activity hours (obtained from Ibarquengoytía et al.²) instead of mean annual air temperature. We find that no model fitted the data well when we included activity time (p values < 0.001). So, we only included mean annual air temperature in our manuscript. Yet, we included the potential implication of higher activity hours in viviparous species to achieve larger sizes than oviparous species into the Discussion, as this is an important consideration.

Comment:

134: the systematic nature of the study requires more information about the source of species' descriptions

Response:

98% of diet information comes from the most recently published database of diet in *Liolaemidae*¹⁷. Therefore, we did not use species' descriptions (as in the original submission) to gather diet information. We did, however, include greater detail about how dietary categories were determined for each species in the database.

Comment:

What is the effect of missing data on the model?

Response:

Our model included 111 species, which represent ~60% of the species represented in the ultrametric tree. As the species included in our model include representatives from the major clades in *Liolaemus*, we consider that missing data are not affecting the model. Further, we reran the phylogenetic path analysis including only 75, 54, or 32 species randomly selected from our database, and we found a consistent pattern. Viviparity indirectly impacts body size through environmental temperature (Figure 2).

Figure 2. Mean annual temperature (MAT) drives body size evolution in *Liolaemus* lizards. To explore the potential effect of missing data we reran the phylogenetic path analysis including 75 (a), 54 (b), or 32 species (c). The average of the best-fitting models of each analysis shows a similar pattern as observed in the analysis with 111 species (Fig. 2a in the main text). Live birth has an indirect effect (through its association with mean annual temperature) on the body size of *Liolaemus* lizards. Continuous arrows (and asterisk) represent significant associations, and thicker arrows indicate stronger effects. Within parentheses we show the confidence interval of each association. The confidence intervals were obtained from 500 bootstrap replicates.

Comment:

148: which priors were used for repeated analyses of the number of character transitions

Response:

We performed a *MuSSCRat* analysis to test whether parity mode (discrete character) is associated with different rates of body size (continuous character) evolution. To test for sensitivity to priors, we ran the analysis using priors of 5, 15 or 25 shifts for both, the discrete and the continuous character. We clarify this in the methods.

Comment:

198: could the percentage of total *Liolaemus* species be provided?

Response:

Added.

Comment:

Figure 2 legend explain boxplot parameters

Response:

Done.

Reviewer #3 (Remarks to the Author):

The authors present a rich dataset including 196 species of *Liolaemus* lizards, their body size, substrate use, diet, and parity mode to test whether parity mode affects body size evolution and diversification. They wonderfully integrate these data with sophisticated, cutting edge, and appropriate phylogenetic comparative methods, and natural history knowledge. They show that viviparity has allowed the lizards to colonize cold habitat niches, although has only modestly affected body size evolution. Although connections between body size, parity mode, diet, and cold environments have long been considered and studied, the authors bring a new perspective through integrating all of these factors in their path analyses, along with possibly the most extensive dataset. This work is certainly worth publishing and makes a great contribution. I have two general comments that the authors should consider, followed by line-specific feedback. The general comments are:

Response:

Thank you for the helpful feedback and kind words.

Comment:

1. There is a slight mismatch in the message in the results and the conclusion. The message in the results seemed to be that viviparity does affect body size evolution, which I had hesitations about. Then the last paragraphs indicates that the effect is not very strong, which I agree with, but seems to be different from how the results are written. I think this is mostly an issue with messaging in the results. The path analysis shows only an indirect effect of parity on body size (really none of the variables have strong effects on body size). The OU modeling shows that the OU1 model, which is really a null model, is not much worse than the OUM and OUMV model. This comment can be addressed simply by making the message that body size is only weakly related to other variables more clear in the results section.

Response:

Thank you for the observation, and we agree this could have been more precise. In our revision, we clarify in the results that the effect of viviparity on body size is indirect. Viviparous species are larger because they inhabit colder habitats than their oviparous relatives (Fig. 2 in the main text).

Comment:

2. Including a wider range of models in the BM/OU modeling and possibly the path analysis may provide more insights around what shapes body size evolution.

a. Currently, the OU modeling includes BM1, OU1, BMS, OUM, OUMV. First two are null, last three are centered on parity mode. I suggest a wider set of models that consider alternative hypotheses for which the authors have data and that they talk about in the discussion. In particular, other adaptive models that might allow the authors to tease apart what is going on are BMS, OUM, and OUMV using diet and substrate (separately). For example, it is possible that an OUM model with diet ends up being the best, and this would suggest that body size evolution is driven by diet and not parity mode. The fact is that currently only a small set of models are considered and one of the null models (OU1) is not much worse than the adaptive models for parity mode. No really viable alternative hypothesis is considered. A possible non-adaptive model that the authors could consider is BMS, OUM, OUMV with different clades identified. The branch colors in Figure 1 suggest that there might be more of clade-specific shifts in body size evolutionary rates.

Response:

We appreciate the suggestions and realize there is value in running evolutionary models using other predictor variables. In our revision, we reran BM1, OU1, BMS, OUM, OUMV models using diet and substrate use. We included the results in the main text and the supplementary data.

Comment:

b. The three models considered in the path analysis are all very similar and great at teasing out how parity might or might not affect body size. Some additional models could be used to understand the effects of the other variables already included in the analysis. One thing that I find striking in the path analysis (Figure 3) is that none of the factors seem to have a very strong effect on body size evolution (only mean annual temperature has a significant effect, but it does not seem very strong).

Response:

In our revision, we ran 15 different models, including all the potential direct and indirect effects of viviparity on body size evolution (Supplementary Figure 2). Consistent with our previous analysis, only mean annual temperature has a significant effect on body size. As the SLOUCH analysis indicates and as the reviewer noted, body size adapts slowly to thermal habitat.

Comment:

Specific comments:

Abstract:

- On line 17, the authors mention that ecological opportunity may “reconfigure phenotypic diversity”, but this idea does not seem to be revisited in the abstract, at least not explicitly. I suspect that this is connected to larger body size in viviparous lizards, which seems less a reconfiguration. I would make that connection more explicit.

Response:

We made an explicit connection between ecological opportunity (colonization of cold habitats by viviparous species) and larger body sizes in our revision.

Comment:

Introduction:

Line 39: Delete “colonize”, as access than colonize give the same meaning here.

Response:

Done.

Comment:

Lines 39-42: This sentence unrealistically simplifies the situation: there certainly are temperate oviparous species as well as plenty of viviparous tropical species of squamates.

Response:

We rephrased the sentence.

Comment:

Line 87: “repeated” (delete “ly”).

Response:

Corrected.

Comment:

Line 93: “another” should be “other”.

Response:

Corrected.

Comment:

Material and Methods:

Lines 120-123: It sounds like air temperature was sampled such that there was one number per species based on one individual from which SVL was collected. I would clarify that this is the case. One could imagine other approaches, like identifying a geographic range based on museum specimens available and taking the average air temperature across all pixels within the range.

Response:

We clarified that body size of adult females was measured in a locality per species and that we gathered data on general air temperature trends for each of those localities.

Comment:

Line 145: Was a 10% burn-in adequate? Was this evaluated in some way?

Response:

Yes, we scrutinized the MCMC runs in Tracer. Specifically, we assessed the effective sample size (ESS>200) for the model itself, as well as key parameters. We added this information to our revision.

Comment:

Line 147: The authors mention key parameters (plural), but then list only a single parameter, the number of rate shifts, for which they tried three different values. I have a couple questions about this. First, were there any other parameters for which priors were required? If so, what were they and how were they treated? Second, Bayesian approaches typically view parameters as distributions, so it is strange to provide a single value (like 5, 10 or 20) to characterize a prior. Was there a distribution to the prior? What were the characteristics of that distribution?

Response:

We added clarification that the number of transitions in the discrete character (parity mode) requires a prior. We repeated the analyses with priors of 5, 15, and 25 transitions. MuSSCRat uses a single value prior on the number of rate shifts. This value is then multiplied by a distribution of the magnitude of rate shifts, which is a lognormal distribution (this distribution is generated internally). So, the single value prior is internally converted into a lognormal distribution that considers the number of rate shifts and their magnitude.

Comment:

Lines 159-164: OU models also have a parameter, alpha, which is the strength of selection to an optimum, but the authors do not mention it. This should be included. Although OUwie also allows for fitting of OUMA models that allow alpha to differ between parity modes, I do not think that is necessary here.

Response:

We specify this in our revision.

Comment:

Lines 204-206: The use of Fischer's C statistic and corresponding p-value is unclear. The authors state that a low p-value indicates that the available evidence rejects the model. What is the null hypothesis of this test? What is it evaluating? Is this a model fit test of some sort?

Response:

Phylogenetic path analysis in the R package phylopath²⁸ provides two statistical approaches to evaluate if the models are a good fit of the data. First, it calculates the *p*-value accompanying the test of the C-statistic. Second, it performs a model selection, which provides the C-statistic information criterion (CIC), the difference in CICc with the best model (deltaCICc), and CICc weights. Using both approaches to decide upon the best model(s) is a standard procedure in phylogenetic path analysis²⁸⁻³⁰. The Fisher's C test³¹ reflects the deviation of the data from the

correlational structure predicted if the model is correct²⁹. Specifically, the correlational structure implied by a set of paths in a causal hypothesis implies a specific set of conditional independencies in the data, where the p -value of a test of correlation should be 0 if the causal hypothesis is correct. The principle is that many causal hypotheses may be consistent with or plausible given the data, but many can also be rejected based on correlations inconsistent with the causal hypothesis. Phylogenetic path analysis identifies these predicted conditional independencies from each causal hypothesis. A p -value lower than 0.05 for these predicted independencies is taken as evidence against the hypothesis and, therefore, the model wouldn't be considered a good fit for the data (*i.e.* it is an implausible causal hypothesis). Based on this, different models have the potential to fit the same data and model selection is then used to identify the best fit among the set of "plausible" path models²⁹. We now provide more details regarding these metrics in the methods section.

Comment:

Results:

Line 230: What the posterior probability represents is unclear. Looking at figure 1, I assume that the boxplots in the top left represent the posterior probability distributions for the rate parameter for oviparous and viviparous species. Does the posterior probability (0.77-0.89) represent the probability that the two rates are different? Whether this is the case or not, this needs to be explained better.

Response:

Thank you for the observation. In our revision, we clarified that posterior probability represents the probability that the rates of body size evolution depend on parity mode.

Comment:

Line 230: I suggest staying in past tense, so "find" should be "found" (and other places).

Response:

Done.

Comment:

Lines 231-233: The headings to supplementary tables 1 and 2 look the same. I suggest modifying the beginning to explain what alternative phylogeny is being used in the second table. Also, looking at these tables (especially supplementary table 1), it is worth considering that the 3rd best model, which suggests no difference in SVL between oviparous and viviparous species, is also not far behind, with a delta of three. This suggests that the support for different body size between the two groups is moderate. Is modeling averaging possible here? It was done with the path analysis?

Response:

We modified the heading for clarity. As we reran the OUwie analyses, the delta valued changed a little bit. Now, delta is 5.6 for the OU model, 0 for the OUM model, and 0.7 for the OUMV model. Therefore, differences in body size between parity modes is supported.

Comment:

Lines 245-246: I suggest more clearly explaining what is meant by the optimal and evolutionary regressions being decoupled. Looking at supplementary table 4 it looks like none of the slopes are significantly different from zero (the SE of the slopes are only slightly lower than the slopes themselves).

Response:

Clarified.

Comment:

Discussion:

Line 286: Including when the Andes began forming would provide useful context for this discussion of dates. Did the Andes predate, coincide with, or postdate the origin of *Liomaemus*?

Response:

This is a good point. Whereas uplift of the Andes began ~100 million years ago, the origin of *Liolaemus* likely occurred ~26 million years ago. We provide this information in the revision.

Comment:

Line 296: Although temperature is a direct predictor of body size and is the only one in the path model, the effect seems somewhat weak (Fig. 3).

Response:

We rephrased the sentence to clarify that effect of environmental temperature on body size is weak.

Comment:

Line 313: Larger species also often feed on a larger range of prey sizes.

Response:

As a result of other changes in the results, we deleted that sentence.

Comment:

Lines 314-315: It seems that the authors have the data to test some of this, for example, an OU model for body size evolution that has different optima for different diets.

Response:

In our revision, we ran additional OUwie analyses by diet and substrate use and incorporated the results in the manuscript as well in the supplementary information.

Comment:

Lines 359-360: I suggest rephrasing this statement to communicate right away that Bergmann's rule is supported in only some groups, and the results from reptiles are very much mixed. I know this appears a few lines later, but the statement in these lines seems too bold.

Response:

We rephrased the sentence.

Comment:

Lines 392-393: I agree with this statement, but feel as though the results of the paper, while accurately describing the results were trying to convince me that viviparity does influence body size evolution (for example, supplementary table 1, in main figure 2).

Response:

We rephrased the sentence to clarity.

Comment:

Line 393: Should it read "...influence body size evolution directly...?"

Response:

We clarified this.

Comment:

Figure 1: This is a great figure, but some basic modifications would make it easier to understand and interpret. The caption should indicate that the boxplots are posterior probability distributions of the Brownian rate parameter (σ), that the branches are colored by rates of evolution, and explain the temperature bar along with the lines connecting to it from the species data. I also suggest that the black scale bar for SVL be broken down a little with some visual guide going across the species data. For example, use two different shades of grey to show 0, 50, and 100 mm points and then use a transparent filled box or a light line (maybe dashed) going down from each point so that it is easier to see how big the SVL of species further down is. Finally, the colors used for omnivore and herbivore are a little difficult to distinguish.

Response:

Following changes in the results, we ended up excluding that figure in our revision.

Comment:

Figure 2: This is another great figure, but the 500 points blend in to form green and purple blobs. I suggest making the points smaller and seeing if there are other ways to make them more distinct.

Response:

Thank you for the suggestion. In our revision, we made the points smaller.

Comment:

Supplementary Table 3: The meaning of the c-statistic and associated p-value should be mentioned here as well.

Response:

Done.

References

1. Pincheira-Donoso, D., Tregenza, T., Witt, M. J. & Hodgson, D. J. The evolution of viviparity opens opportunities for lizard radiation but drives it into a climatic cul-de-sac. *Glob. Ecol. Biogeogr.* **22**, 857–867 (2013).
2. Ibargüengoytía, N. R. *et al.* Looking at the past to infer into the future: Thermal traits track environmental change in Liolaemidae*. *Evolution* **75**, 2348–2370 (2021).
3. Cruz, F. B. *et al.* The role of climate and maternal manipulation in determining and maintaining reproductive mode in *Liolaemus* lizards. *J. Zool.* **317**, 101–113 (2022).
4. Cabrera, M. P., Scrocchi, G. J. & Cruz, F. B. Sexual size dimorphism and allometry in *Liolaemus* of the *L. laurenti* group (Sauria: Liolaemidae): Morphologic lability in a clade of lizards with different reproductive modes. *Zool. Anz.* **252**, 299–306 (2013).
5. Etheridge, R. A review of lizards of the *Liolaemus wiegmanni* group (Squamata, Iguania, Tropicuridae), and a history of morphological change in the sand-dwelling species. *Herpetol. Monogr.* 293–352 (2000).
6. Meiri, S. Evolution and ecology of lizard body sizes. *Glob. Ecol. Biogeogr.* **17**, 724–734 (2008).
7. Tulli, M. J., Cruz, F. B., Herrel, A., Vanhooydonck, B. & Abdala, V. The interplay between claw morphology and microhabitat use in neotropical iguanian lizards. *Zoology* **112**, 379–392 (2009).
8. Abdala, C. S., Sebastián Quinteros, A., Arias, F., Portelli, S. & Palavecino, A. A new species of the *Liolaemus darwini* group (Iguania: Liolaemidae) from Salta Province, Argentina. *Zootaxa* 26 (2011).
9. Jara, M. *et al.* The macroecology of chemical communication in lizards: Do climatic factors drive the evolution of signalling glands? *Evol. Biol.* **45**, 259–267 (2018).
10. Lobo, F. & Laurent, R. F. Un nouveau *Liolaemus* Andin (Tropicuridae). *Revue fr. Aquariol.* **22**, 107–116 (1995).
11. Cruz, F. B., Fitzgerald, L. A., Espinoza, R. E. & Schulte, J. A. The importance of phylogenetic scale in tests of Bergmann's and Rapoport's rules: Lessons from a clade of South American lizards. *J. Evol. Biol.* **18**, 1559–1574 (2005).
12. Abdala, C. S. & Gómez, J. M. D. A new species of the *Liolaemus darwini* group (Iguania: Liolaemidae) from Catamarca Province, Argentina. *Zootaxa* 21–33 (2006).
13. Tulli, M. J., Cruz, F. B., Kohlsdorf, T. & Abdala, V. When a general morphology allows many habitat uses. *Integr. Zool.* **11**, 483–499 (2016).
14. Tulli, M. J. & Cruz, F. B. Are the number and size of scales in *Liolaemus* lizards driven by climate? *Integr. Zool.* **13**, 579–594 (2018).
15. Ruiz-Monachesi, M. R., Cruz, F. B., Valdecantos, S. & Labra, A. Unravelling associations among chemosensory system components in *Liolaemus* lizards. *J. Zool.* **312**, 148–157 (2020).
16. Ruiz, S. *et al.* Diversification and geological history of the *Liolaemus ornatus* group (Squamata: Iguania) of Argentina including the recognition of a new species. *Zool. Anz.* **292**, 126–138 (2021).

17. Ocampo, M., Pincheira-Donoso, D., Sayol, F. & Rios, R. S. Evolutionary transitions in diet influence the exceptional diversification of a lizard adaptive radiation. *BMC Ecol. Evol.* **22**, 1–10 (2022).
18. Espinoza, R. E., Wiens, J. J. & Tracy, C. R. Recurrent evolution of herbivory in small, cold-climate lizards: Breaking the ecophysiological rules of reptilian herbivory. *Proc. Natl. Acad. Sci. U.S.A.* **101**, 16819–16824 (2004).
19. Meiri, S. Traits of lizards of the world: Variation around a successful evolutionary design. *Glob. Ecol. Biogeogr.* **27**, 1168–1172 (2018).
20. Shine, R. Evolution of an evolutionary hypothesis: A history of changing ideas about the adaptive significance of viviparity in reptiles. *J. Herpetol.* **48**, 147–161 (2014).
21. Ma, L., Buckley, L. B., Huey, R. B. & Wei-Guo, D. A global test of the cold-climate hypothesis for the evolution of viviparity of squamate reptiles. *Glob. Ecol. Biogeogr.* 1–11 (2018) doi:10.1111/geb.12730.
22. Zimin, A. *et al.* A global analysis of viviparity in squamates highlights its prevalence in cold climates. *Glob. Ecol. Biogeogr.* (2022).
23. Ashton, K. G., Tracy, M. C. & De Queiroz, A. Is Bergmann’s rule valid for mammals? *Am. Nat.* **156**, 390–415 (2000).
24. Ashton, K. G. & Feldman, C. R. Bergmann’s rule in nonavian reptiles: Turtles follow it, lizards and snakes reverse it. *Evolution* **57**, 1151–1163 (2003).
25. McQueen, A. *et al.* Thermal adaptation best explains Bergmann’s and Allen’s Rules across ecologically diverse shorebirds. *Nat. Commun.* 2022 13:1 **13**, 1–12 (2022).
26. Toyama, K. S. & Boccia, C. K. Bergmann’s rule in *Microlophus* lizards: testing for latitudinal and climatic gradients of body size. Preprint at <https://doi.org/10.1101/2022.01.18.476846> (2022).
27. Mull, C. G., Pennell, M. W., Yopak, K. E. & Dulvy, N. K. Maternal investment evolves with larger body size and higher diversification rate in sharks and rays. Preprint at <https://doi.org/10.1101/2022.01.05.475057> (2022).
28. van der Bijl, W. phylopath: Easy phylogenetic path analysis in R. *PeerJ* **2018**, e4718 (2018).
29. Von Hardenberg, A. & Gonzalez-Voyer, A. Disentangling evolutionary cause-effect relationships with phylogenetic confirmatory path analysis. *Evolution* **67**, 378–387 (2013).
30. Gonzalez-Voyer, A. & Von Hardenberg, A. An Introduction to phylogenetic path analysis. in *Modern Phylogenetic Comparative Methods and Their Application in Evolutionary Biology* (ed. Garamszegi, L. Z.) 201–229 (Springer, 2014).
31. Shipley, B. *Cause and correlation in biology: A user’s guide to path analysis, structural equations and causal inference.* (Cambridge University Press, Cambridge, England, 2002).

Reviewers' Comments:

Reviewer #1:

Remarks to the Author:

I read the manuscript "Viviparity imparts a macroevolutionary signature of ecological opportunity in *Liolaemus* lizards" by Dominguez-Guerrero and colleagues. As mentioned in the first round I found the manuscript interesting and well written. In this second round, many aspects were clarified and my impression is that the manuscript was improved. However, I still have some concerns. The most important one is regarding the data base and the SVL data for the *Liolaemus* females. On my first review, I mentioned the authors that I could share some information with them, without any other interest. Now, one of the authors asked me for SVL data from 45 species; I send the data considering these data would be of interest and useful for the manuscript. After observing figure 1, I found that the optimal body size for oviparous species was really small. I have to say that the value of the optimal size may vary from the raw data for several reasons that I do not know in detail. In any case, I took a look to the data set in the Excel file provided by the authors. To my surprise, several of the data I offered the authors were no used, but the fact that really surprised me is that for nine species (*Liolaemus canqueli*, *L. cuyanus*, *L. fitzingerii*, *L. goetschi*, *L. koslowskyi*, *L. olongasta*, *L. pseudoanomalus*, *L. salinicola* and *L. xanthoviridis*) the authors chose to use data from other sources where SVL was between 4 and 15 mm smaller. I do not know the reason for this, especially after the author that requested the data insisted several times. It is possible that the data I sent will be used in a different manuscript, but it really call my attention. For this I need an explanation.

Other concerns.

Lines 187-194. About the colonization of cold climates. Why did the authors ignored the Díaz Gómez papers? It is possible that the trees these authors used are not time calibrated; however, the maps would be very useful to read and based on the knowledge that one of the authors might have, since he was borne in Chile, these maps could give an idea on what were the areas of ancestry and dispersion. I am listing the papers again. Additionally, was Patagonia cold 20 million years ago?

Díaz Gómez JM (2011) Estimating Ancestral Ranges: Testing Methods with a Clade of Neotropical Lizards (Iguania: *Liolaemidae*). PLoS ONE 6(10): e26412.

doi:10.1371/journal.pone.0026412

Díaz Gómez, J.M. (2009), Historical biogeography of *Phymaturus* (Iguania: *Liolaemidae*) from Andean and patagonian South America. *Zoologica Scripta*, 38: 1-

7. <https://doi.org/10.1111/j.1463-6409.2008.00357.x>

Díaz Gómez, Juan Manuel;; Lobo, Fernando; 2006 Historical Biogeography of a clade of *Liolaemus* (Iguania: *Liolaemidae*) based on ancestral areas and dispersal-vicariance analysis (DIVA). *PAPEIS AVULSOS DE ZOOLOGIA.*; Lugar: San Pablo, Brasil; 46 p. 261 – 274

Lines 232-235. The very same author that asked me for the SVL data from females, commented me about Cabrera et al (2013) paper from which I am a coauthor (Cabrera, MP.; G. Scrocchi and FB. Cruz. 2013. Sexual size dimorphism and allometry in *Liolaemus* of the *L. laurenti* group (Sauria: *Liolaemidae*): Morphologic lability in a clade of lizards with different reproductive modes. *Zoologischer Anzeiger* 252: 299-306.). This paper contains information about sexual size dimorphism in *Liolaemus* of the *L. laurenti* clade and also information about the lack of difference in fecundity between viviparous and oviparous species there. My point is that this issue was considered, but insufficiently documented.

Line 255. Just a comment. Rocky habitats despite the interesting association te authors noted, it is a type of habitat and substrate where an ovipoarous species would have difficulties to build or dig a nest, as oviparous *Liolaemus* do. In any case, the are exceptions, the *L. ornatus* clade occur in sandy habitats and they are viviparous.

Line 258. It has to be said that both species (*Liolaemus sarmientoi* and *L. magellanicus*) are viviparous, despite the difference in other aspects, such as thermoregulation and microhabitat use. Finally Line 313. The authors are ignoring that *Liolaemus* from the *L. laurenti* clade and *Phymaturus*

lizards do not show SSD, please consider this.

Well, in my opinion the manuscript is, as I said before, very interesting, novel and well written. Clarifying the aspects, I mentioned above, I am pretty sure the next version will be closer for publishing. As you may notice, I am not a native English speaker, for this reason I am not correcting grammar or orthography.

Reviewer #2:

Remarks to the Author:

The authors have addressed my comments satisfactorily.

Reviewer #3:

Remarks to the Author:

I have now reviewed the revised manuscript and wish to reiterate that this is a great study of parity mode evolution and body size in *Liolaemus* lizards. I enjoyed reading it, felt it was clear and compelling, and integrated a number of datasets using cutting-edge phylogenetic comparative stats. The authors mostly addressed the concerns that I brought up originally. However, I think that they misunderstood what I suggested in terms of the OU/BM modeling. This now appears on lines 107-111 and supplementary tables 1-6. I appreciate that the authors added OU and BM models for different diets and parity modes, but how this was implemented was not quite what I had suggested. My apologies for anything that was not clear in my previous comments. The models should all be considered together in a single table so that alternative hypotheses can be tested. Currently, in each table separately, one can say whether a model of parity or diet or habitat out performed some null models. If the models are all combined into a single table, then one can determine the strength of evidence for models that consider parity versus models that consider diet versus those that consider habitat versus null models. This would be far more informative, as one could determine if SVL is more strongly shaped by one or another of these factors. For example, one might find that diet really dictates SVL much more strongly than parity, or vice versa. The authors should present a single table with BM1, OU1, BMS(parity), OUM(parity), OUMV(parity), BMS(diet), OUM(diet), OUMV(diet), BMS(habitat), OUM(habitat), OUMV(habitat) and rank all those models using delta AICc values. I'll note that this is not just a semantic comment about number of tables, but about the comparison of the full set of models. I believe that this needs to be addressed prior to publication.

I had a few minor comments as well, really just typos/formatting:

L124: "influences"

L454: "is" and "t1/2" are transposed, their order should be switched for meaning.

Supplementary Table 8: This table looks like it has some formatting issues. The column headings do not line up with the data, and for models 13, 8, and 14 (rows 2-5), the columns are shifted even more so that they do not line up with the other models.

REVIEWER COMMENTS

Reviewer #1 (Remarks to the Author):

Comment:

I read the manuscript “Viviparity imparts a macroevolutionary signature of ecological opportunity in *Liolaemus* lizards” by Dominguez-Guerrero and colleagues. As mentioned in the first round I found the manuscript interesting and well written. In this second round, many aspects were clarified and my impression is that the manuscript was improved. However, I still have some concerns. The most important one is regarding the data base and the SVL data for the *Liolaemus* females. On my first review, I mentioned the authors that I could share some information with them, without any other interest. Now, one of the authors asked me for SVL data from 45 species; I send the data considering these data would be of interest and useful for the manuscript. After observing figure 1, I found that the optimal body size for oviparous species was really small. I have to say that the value of the optimal size may vary from the raw data for several reasons that I do not know in detail. In any case, I took a look to the data set in the Excel file provided by the authors. To my surprise, several of the data I offered the authors were not used, but the fact that really surprised me is that for nine species (*Liolaemus canqueli*, *L. cuyanus*, *L. fitzingerii*, *L. goetschi*, *L. koslowskyi*, *L. olongasta*, *L. pseudoanomalus*, *L. salinicola* and *L. xanthoviridis*) the authors chose to use data from other sources where SVL was between 4 and 15 mm smaller. I do not know the reason for this, especially after the author that requested the data insisted several times. It is possible that the data I sent will be used in a different manuscript, but it really call my attention. For this I need an explanation.

Response:

We thank the reviewer for sharing body size data of 45 *Liolaemus* species (Table 1). These data, derived from a Supplementary Information of a published paper (Cruz *et al.*, 2022)¹, were considered and useful to build our body size database. As the reviewer observed optimal body size showed in the figure 1 varies from raw data. It is because they are different metrics; whereas optimal body size is an inference from an adaptative Ornstein-Uhlenbeck model-fitting procedure (as it is described in the caption of the Fig 1 and the Methods section *Modeling stabilizing selection under Ornstein-Uhlenbeck models*), raw body size is the average of the snout-vent length measured of adult females. We do apologize for any confusion this may have caused.

While we were able to get data from 45 species from Cruz *et al.* (2022) not all data could be used in our study. As we described in Methods (section Body size (SVL)), and in our previous Response to Reviewers document, we followed specific criteria to select appropriate mean body size information of adult females for our dataset. One criterion was that when different papers reported mean SVL of adult females for the same species, we chose the paper with the highest sample size. Therefore, we compared sample size of the species included in Cruz *et al.* (2022) with other sources. Following this criterion, we included body size of eight species (*L. abaucan*, *L. calchaqui*, *L. darwini*, *L. irregularis*, *L. kriegi*, *L. laurenti*, *L. lavillai*, and *L. ornatus*) because the sample size was larger compared with other sources. We did not use body size of *L.*

casamiquelai, *L. purul*, and *L. scrocchii* because these species are not represented in the ultrametric tree² and therefore they cannot be included in the evolutionary analyses. We did not include body size of *L. crepuscularis* and *L. grosseorum* because data of these species comes from Cabrera et al., (2013)³ and in that study body size was estimated using the larger third of the individuals rather than the mean SVL of adult females. We did not include body size data for *L. baguali*, *L. canqueli*, *L. donobarrosi*, *L. goetschi*, *L. kingii*, *L. kolengh*, *L. martorii*, *L. petrophilus*, *L. rothi*, *L. ruibali*, *L. salinicola*, *L. xanthoviridis*, and *L. zullyae* because sample size is not specified whereas in other sources it is. Lastly, we did not include body size data of *L. albiceps*, *L. bibronii*, *L. chacoensis*, *L. coeruleus*, *L. cuyanus*, *L. elongatus*, *L. espinozai*, *L. famatinae*, *L. fitzingerii*, *L. gallordi*, *L. koslowskyi*, *L. multicolor*, *L. multimaculatus*, *L. olongasta*, *L. poecilochromus*, *L. pseudoanomalus*, *L. quilmes*, *L. riojanus*, and *L. scapularis* because other sources afford higher sample sizes for these species.

As we believe that the Reviewer's concern is that using different sources of body size data could impact our findings, we re-ran the OUwie analysis (evolution of body size according to parity mode) including data of all species (except the three species that are not included in the ultrametric tree) from Cruz *et al.* (2022). We found a consistent pattern: viviparous species evolve a larger optimal body size ($\theta = 66.4$ mm) than their oviparous relatives ($\theta = 56.6$ mm). Therefore, using different sources of body size data in the evolutionary analyses did not impact our results.

Table 1. Mean snout-vent length of adult females provided by Cruz et al. (2022).

Species	mean SVL ♀	standard deviation
L. abaucan	53.52	1.75
L. albiceps	71.52	5.61
L. baguali	81.79	2.65
L. bibronii	55.98	1.53
L. calchaqui	57.46	5.6
L. canqueli	88.7	2.15
L. casamiquelai	63	1.34
L. chacoensis	46.5	0.98
L. coeruleus	63.27	1.15
L. crepuscularis	58.83	5.12
L. cuyanus	78.8	2.38
L. darwinii	55.66	2.81
L. donosobarrosi	55.33	5.62
L. elongatus	63.13	2.18
L. espinozai	55.3	5.07
L. famatinae	55.26	0.76
L. fitzingerii	89.92	2.44
L. gallardoii	74.04	1.15
L. goestchi	71.95	4.37
L. grosseorum	49.38	1.55
L. irregularis	71.49	8.49
L. kingii	74.98	2.41

L. kolengh	56.66	0.88
L. koslowskyi	59.7	3.78
L. kriegi	98.77	3.26
L. laurenti	52.2	3.67
L. lavillai	54.72	6.14
L. martorii	60.38	2.49
L. multicolor	63.3	2.31
L. multimaculatus	60.49	2.82
L. olongasta	54.51	1.91
L. ornatus	59.9	3.54
L. petrophilus	78.46	2.65
L. poecilochromus	63.56	1.16
L. pseudoanomalous	58.31	6.57
L. purul	65.5	2.16
L. quilmes	53.42	2.06
L. riojanus	48.73	3.02
L. rothi	78.7	9.81
L. ruibali	55.26	0.76
L. salinicola	57.13	3.29
L. scapularis	57.54	4.44
L. scrocchii	86.98	2.11
L. xanthoviridis	84.17	2.9
L. zullyae	66.5	3.12

Table 2. Summary of five model fits for the body size (snout-to-vent length; SVL) data across 500 stochastic character maps of parity mode through the maximum clade credibility tree. For this analysis, we included body size data of 42 *Liolaemus* species estimated by Cruz *et al.* (2022). A brief description of each model is as follows: (1) BM1: a single-rate model of stochastic trait evolution (i.e., rate constrained to be the same for viviparous and oviparous species), (2) BMS: a two-rate model of stochastic trait evolution (i.e., separate rates for viviparous and oviparous species), (3) OU1: a single phenotypic optimum (i.e., shared optimum for viviparous and oviparous species) and a single-rate model, (4) OUM: a two optima, single-rate model, and (5) OUMV: a two optima, two-rate model. This analysis was conducted with SVL data from 135 species (59 oviparous and 76 viviparous). For the best-fitting model (OUM), we provide the rate of trait evolution (σ^2), the strength of selection (α), and the phenotypic optimum (θ).

Model	$\Delta AICc$	Weight	Rate of trait evolution (σ^2) and alpha (α)	Phenotypic optimum (θ)
BM1	29.9	<0.001	-	-
BMS	30.9	<0.001	-	-
OU1	5.3	0.05	-	-
OUM	0	0.70	$\sigma^2=0.0018$ and $\alpha=0.148$ for all species	56.6 mm SVL for oviparous species and 66.4 mm SVL for viviparous species
OUMV	2.1	0.25		

Comment:

Other concerns.

Lines 187-194. About the colonization of cold climates. Why did the authors ignore the Díaz Gómez papers? It is possible that the trees these authors used are not time calibrated; however, the maps would be very useful to read and based on the knowledge that one of the authors might have, since he was born in Chile, these maps could give an idea on what were the areas of ancestry and dispersion. I am listing the papers again. Additionally, was Patagonia cold 20 million years ago?

Díaz Gómez JM (2011) Estimating Ancestral Ranges: Testing Methods with a Clade of Neotropical Lizards (Iguania: Liolaemidae). PLoS ONE 6(10): e26412.

doi:10.1371/journal.pone.0026412

Díaz Gómez, J.M. (2009), Historical biogeography of Phymaturus (Iguania: Liolaemidae) from Andean and Patagonian South America. Zoologica Scripta, 38: 1-

7. <https://doi.org/10.1111/j.1463-6409.2008.00357.x>

Díaz Gómez, Juan Manuel;; Lobo, Fernando; 2006 Historical Biogeography of a clade of Liolaemus (Iguania: Liolaemidae) based on ancestral areas and dispersal-vicariance analysis (DIVA). PAPEIS AVULSOS DE ZOOLOGIA.; Lugar: San Pablo, Brasil; 46 p. 261 – 274

Response:

Thank you for the suggestion. We incorporated in the manuscript the hypotheses of Díaz Gómez about ancestral origin and dispersion of *Liolaemus*. Yes, South American regions, including Patagonia, have progressively cooled during the last 20 million years⁴.

Comment:

Lines 232-235. The very same author that asked me for the SVL data from females, commented me about Cabrera et al (2013) paper from which I am a coauthor (Cabrera, MP.; G. Scrocchi and FB. Cruz. 2013. Sexual size dimorphism and allometry in *Liolaemus* of the *L. laurenti* group (Sauria: Liolaemidae): Morphologic lability in a clade of lizards with different reproductive modes. Zoologischer Anzeiger 252: 299-306.). This paper contains information about sexual size dimorphism in *Liolaemus* of the *L. laurenti* clade and also information about the lack of difference in fecundity between viviparous and oviparous species there. My point is that this issue was considered, but insufficiently documented.

Response:

Thank you for the suggestion, we expanded our explanation about the potential benefit of larger sizes in viviparous females (compared with their oviparous relatives) in terms of fecundity and abdominal space to developing embryos as suggested in Cabrera et al. (2013)³.

Comment:

Line 255. Just a comment. Rocky habitats despite the interesting association the authors noted, it is a type of habitat and substrate where an oviparous species would have difficulties to build or

dig a nest, as oviparous *Liolaemus* do. In any case, there are exceptions, the *L. ornatus* clade occurs in sandy habitats and they are viviparous.

Response:

It is a good observation that we incorporated in the manuscript.

Comment:

Line 258. It has to be said that both species (*Liolaemus sarmientoi* and *L. magellanicus*) are viviparous, despite the difference in other aspects, such as thermoregulation and microhabitat use.

Response:

We clarified that both species are viviparous.

Comment:

Finally Line 313. The authors are ignoring that *Liolaemus* from the *L. laurenti* clade and *Phymaturus* lizards do not show SSD, please consider this.

Response:

Good point, we modify that sentence to say that *Liolaemus* species can exhibit sexual dimorphism, thus not affirming all species exhibit sexual dimorphism. As we focused our work in *Liolaemus*, lack of sexual dimorphism in *Phymaturus* was not relevant for our body size compilation.

Comment:

Well, in my opinion the manuscript is, as I said before, very interesting, novel and well written. Clarifying the aspects, I mentioned above, I am pretty sure the next version will be closer for publishing. As you may notice, I am not a native English speaker, for this reason I am not correcting grammar or orthography.

Response:

Thank you for all the suggestions, and we greatly appreciate your kind words and feedback.

Reviewer #2 (Remarks to the Author):

Comment:

The authors have addressed my comments satisfactorily.

Response:

Thank you for all the positive feedback.

Reviewer #3 (Remarks to the Author):

Comment:

I have now reviewed the revised manuscript and wish to reiterate that this is a great study of parity mode evolution and body size in *Liolaemus* lizards. I enjoyed reading it, felt it was clear and compelling, and integrated a number of datasets using cutting-edge phylogenetic comparative stats. The authors mostly addressed the concerns that I brought up originally. However, I think that they misunderstood what I suggested in terms of the OU/BM modeling. This now appears on lines 107-111 and supplementary tables 1-6. I appreciate that the authors added OU and BM models for different diets and parity modes, but how this was implemented was not quite what I had suggested. My apologies for anything that was not clear in my previous comments. The models should all be considered together in a single table so that alternative hypotheses can be tested. Currently, in each table separately, one can say whether a model of parity or diet or habitat out performed some null models. If the models are all combined into a single table, then one can determine the strength of evidence for models that consider parity versus models that consider diet versus those that consider habitat versus null models. This would be far more informative, as one could determine if SVL is more strongly shaped by one or another of these factors. For example, one might find that diet really dictates SVL much more strongly than parity, or vice versa. The authors should present a single table with BM1, OU1, BMS(parity), OUM(parity), OUMV(parity), BMS(diet), OUM(diet), OUMV(diet), BMS(habitat), OUM(habitat), OUMV(habitat) and rank all those models using delta AICc values. I'll note that this is not just a semantic comment about number of tables, but about the comparison of the full set of models. I believe that this needs to be addressed prior to publication.

Response:

We appreciate the reviewer suggestion and we apologize that we misunderstood the original comment. Previously, we performed each analysis with a different number of species ($n=133$ for parity mode, $n=119$ for diet, and $n=123$ for substrate use) and we decide to not collate these delta AICc values in a same table as it could be affected by sample size. Now that we understand the reviewer's point better, we reran the OUwie analyses for the 111 *Liolaemus* species for which we know their parity mode, diet, and substrate use. We found that both delta AICc and AIC weight support OUM (parity mode) as the best-fitting model. We incorporated the results into a single table in the supplementary information as well in the main text.

Comment:

I had a few minor comments as well, really just typos/formatting:
L124: "influences"

Response:

Corrected.

Comment:

L454: "is" and "t1/2" are transposed, their order should be switched for meaning.

Response:

Switched.

Comment:

Supplementary Table 8: This table looks like it has some formatting issues. The column headings do not line up with the data, and for models 13, 8, and 14 (rows 2-5), the columns are shifted even more so that they do not line up with the other models.

Response:

We corrected the formatting issue. Thank you for all the kind and valuable suggestions.

References

1. Cruz, F. B. *et al.* The role of climate and maternal manipulation in determining and maintaining reproductive mode in *Liolaemus* lizards. *J. Zool.* **317**, 101–113 (2022).
2. Esquerré, D., Brennan, I. G., Catullo, R. A., Torres-Pérez, F. & Keogh, J. S. How mountains shape biodiversity: The role of the Andes in biogeography, diversification, and reproductive biology in South America's most species-rich lizard radiation (Squamata: Liolaemidae). *Evolution* **73**, 214–230 (2019).
3. Cabrera, M. P., Scrocchi, G. J. & Cruz, F. B. Sexual size dimorphism and allometry in *Liolaemus* of the *L. laurenti* group (Sauria: Liolaemidae): Morphologic lability in a clade of lizards with different reproductive modes. *Zool. Anz.* **252**, 299–306 (2013).
4. Skeels, A., Esquerré, D., Lipsky, D., Pellissier, L. & Boschman, L. M. Elevational Goldilocks zone underlies the exceptional diversity of a large lizard radiation (*Liolaemus*; Liolaemidae). *Evolution* **77**, 2672–2686 (2023).

Reviewers' Comments:

Reviewer #1:

Remarks to the Author:

I have read the new version of the manuscript "Viviparity imparts a macroevolutionary signature of ecological opportunity in Liolaemus lizards" by Dominguez-Guerrero and colleagues. After the new modifications and the attention received to my major concerns, I consider the present version of the manuscript should be accepted for publication in Nature Communications.
I congratulate the authors for the great job.

Reviewer #3:

Remarks to the Author:

The authors have now addressed all of my comments. It's a great paper!